# Explainable AI-based analysis of human pancreas sections identifies traits of type 2 diabetes

Lukas Klein[1,2,3,22,24], Sebastian Ziegler[3,4,24], Felicia Gerst[5,6,7,24], Yanni Morgenroth[7,8,9,24], Karol Gotkowski[3,4], Eyke Schöniger[7,8,9], Martin Heni [10,11], Nicole Kipke[7,8,9,12], Daniela Friedland[7,8,9], Annika Seiler[7,8,9], Ellen Geibelt[13], Hajime Yamazaki[14,15], Hans-Ulrich Häring[5,6,7], Silvia Wagner[16], Silvio Nadalin[16], Alfred Königsrainer[16], Andre L. Mihaljevic[16], Daniel Hartmann[16], Falko Fend [16,17], Daniela Aust[18], Jürgen Weitz[7,9,12], Reiner Jumpertz-von Schwartzenberg [5,6,7], Marius Distler[7,9,12], Klaus Maier-Hein[3,4,19], Andreas L. Birkenfeld [5,6,7], Susanne Ullrich[5,6,7], Paul F. Jäger[1,3,23,25] ✉, Fabian Isensee [3,4,25] ✉, Michele Solimena [7,8,9,25] ✉ & Robert Wagner [7,20,21,25] ✉

Type 2 diabetes (T2D) is a chronic disease currently affecting around 500 million people worldwide with often severe health consequences. Yet, histopathological analyses are still inadequate to infer the glycaemic state of a person based on morphological alterations linked to impaired insulin secretion and $\beta$-cell failure in T2D. Giga-pixel microscopy can capture subtle morphological changes, but data complexity exceeds human analysis capabilities. In response, we generate a dataset of pancreas whole-slide images from living donors with multiple chromogenic and multiplex immunofluorescence stainings and train deep learning models to predict the T2D status. Using explainable AI, we make the learned relationships interpretable, quantify them as biomarkers, and assess their association with T2D. Remarkably, the highest prediction performance is achieved by simultaneously focusing on islet $\alpha$- and $\delta$-cells and neuronal axons, alongside subtle pancreatic alterations in T2D donors such as larger adipocyte clusters, altered islet-adipocyte proximity and smaller islets. This data-driven approach provides a foundation for future research into relevant diagnostic and therapeutic targets, refining several hypotheses regarding tissue alterations associated with T2D.

Based on the current WHO classification of diabetes, over 90% of all persons with the condition fall into the category defined as type 2 diabetes (T2D). T2D is a major global health issue, affecting millions and placing a significant burden on healthcare systems[1]. Pathophysiologic drivers of T2D are insulin resistance and impaired insulin

secretion, with dysfunction of pancreatic islet $\beta$-cells being the key feature. Despite extensive research, the exact etiology and nature of $\beta$-cell failure in the pathophysiology of T2D remain unclear. $\beta$-cells reside in specialized micro-organs, the islets of Langerhans, within the pancreas, but accessing human pancreatic tissue in living subjects is

A full list of affiliations appears at the end of the paper. ✉e-mail: paulfjaeger@icloud.com; f.isensee@dkfz.de; michele.solimena@tu-dresden.de; robert.wagner@uni-duesseldorf.de

challenging due to anatomical constraints and the risks associated with pancreatic biopsy or tissue resection[2]. Consequently, the identification of morphological islet changes associated with T2D has relied mainly on the histopathological analysis of specimens from deceased donors. Such studies pointed to several structural alterations, such as amyloid deposition[3,4], reduced $\beta$-cell mass with decreased $\beta$-cell/$\alpha$-cell ratio[5,6], increased number of islet resident macrophages[7,8] and fibrosis[9,10], but several of these findings remain controversial[11–13]. Given the high degree of inter-individual variability, none of these traits is sufficient for pathologists to discriminate with a high degree of certainty whether a pancreatic specimen belongs to a subject with or without T2D. While such discriminative power would not have diagnostic relevance, it could pave the way to uncover yet unknown structural hallmarks of the disease and thereby provide further insight into its pathogenesis. This knowledge could, in turn, be leveraged to improve the prevention and treatment of diabetes.

Recent advances in Machine Learning (ML) have garnered significant attention in the field of computational pathology due to their ability to analyze large volumes of high-dimensional whole-slide images (WSIs) with efficacy comparable to expert pathologists[14]. For example, deep learning (DL) technologies are employed for prostate cancer grading[14], cell segmentation[15], gene expression prediction[16], and the discovery of new morphological biomarkers in breast cancer histopathology[17]. However, most DL applications in histopathology focus on oncological diseases, where distinct cancer cells are identifiable in tissue samples. In contrast, pancreatic tissue lacks clear quantifiable features that unequivocally differ between individuals with and without T2D.

Hence, our goal was the identification of histologic biomarkers allowing us to distinguish a pancreatic specimen from a patient with or without T2D and thus gain novel insights into disease mechanisms linked to, or arising from, morphological alterations of the pancreatic tissue. For that purpose, we introduced a data-driven approach combining state-of-the-art DL with explainable artificial intelligence (XAI). XAI is often employed to enhance model trustworthiness to verify that a model is making decisions based on relevant features, however, this work leverages XAI for scientific discovery purposes. While DL models could find patterns in data that are infeasible to detect by traditional approaches, XAI methods subsequently rendered the learned relationships human-understandable, revealing regions of interest (ROIs) associated with the presence of T2D. Given the limited feasibility of manual analyses of large amounts of XAI results, we quantified the attention to the ROIs and subsequently computed specific histologic biomarkers for the most important ones. At last, we analyzed these biomarkers in combination with clinical patient data using statistical models. Besides the integration of vast amounts of diverse WSIs, our approach offered the advantage that we did not bias ourselves to prior assumptions regarding T2D, as the XAI application could also uncover unanticipated findings, leading to the formulation of new hypotheses about the condition.

## Results

### A unique dataset of IHC and mIF stained pancreas sections
DL models that can reliably distinguish patients with T2D from other patients are a prerequisite for gaining new insights into T2D through XAI methods. By uncovering the patterns used by the DL models for prediction, we aim for a better understanding of the disease.

To this end, we first collected a dataset by applying both single chromogenic immunohistochemistry (IHC) and multiplex immunofluorescence (mIF) techniques to immunostain pancreas sections from 100 metabolically phenotyped living donors with ($n = 35$) or without ($n = 65$) T2D who underwent pancreatectomy for various pancreatic disorders[18–22] at two academic centers. Sections were immunostained for glucagon, insulin, and somatostatin as markers of islet $\alpha$-, $\beta$- and $\delta$-cells, respectively, as well as PECAM1 for endothelial cells to visualize islet vascularization, perilipin 1 for adipocytes to assess intrapancreatic steatosis and tubulin beta 3 for neuronal axons. In the case of IHC stainings, each marker was detected individually in serial sections counterstained with hematoxylin. In the case of mIF stainings, serial sections were co-labeled for several markers, including DAPI and primary antibodies either against glucagon, somatostatin, and tubulin beta 3 (Staining set 1), or against insulin, PECAM1, and perilipin 1 (Staining set 2) (Fig. 1A and "Method" Section). Overall, eight images per patient were acquired. The dataset is publicly available.

### Classification of diabetes from whole-slide images of pancreatic tissue sections
Subsequent brightfield and fluorescence acquired whole slide images (WSIs) were used to train DL models distinguishing donors with or without T2D (Fig. 1b). Patients were split into a training set ($n = 75$) and a test set ($n = 25$), with the test set used only for the final model evaluation. During model development, we employed a cross validation where 15 different training ($n = 60$ each) and validation ($n = 15$ each) splits were created to avoid overfitting on single validation sets. Final test set predictions were obtained by ensembling the 15 individual models, i.e., averaging the individual predictions.

We trained DL models for each of the six IHC stainings, while on the mIF data, we trained DL models for each of the two staining sets (Fig. 2). Each of our DL models consisted of two components, the pretrained feature extractor, and the Multiple Instance Learning (MIL)[23] classifier. Generally, due to their size, each WSI was first divided into smaller-sized squares (patches) and each patch was encoded by the feature extractor. Subsequently, all encodings belonging to one WSI were combined and fed to the MIL classifier, which learned to distinguish between T2D and non-diabetic patients based on the combined patch encodings. By doing so, not only small local features from individual patches but also larger structures spanning multiple patches were available to the model.

We compared the classification performance for different pretraining approaches on the brightfield WSIs. The ImageNet21k pre-trained[24] Vision Transformer as feature extractor combined with the CLAM model architecture[25] as classifier delivered the best average prediction performance ($AUROC = 0.833$; Fig. 2a and Supplementary Table 1). Across all stainings except for PECAM1 and somatostatin, there was a significant increase in AUROC when using the ImageNet21k pre-trained encoder. Of note, the IHC staining resulting in the best prediction performance was tubulin beta 3 ($AUROC = 0.895$), while the three islet-typical labels, i.e., insulin, glucagon, and somatostatin showed similar AUROC values ($AUROC = 0.842$; Fig. 2a and Supplementary Table 1).

On the mIF WSIs, we compared three different color-channel representations: "RGB", "channel-wise", and "channel-wise average" (see subsection 4.3). In short, "RGB" treated the three stainings in a staining set as rgb color channels processing all channels simultaneously, while the two "channel-wise" representations processed each channel individually and either appended the resulting encodings ("channel-wise") or averaged them ("channel-wise average"). The latter benefited from co-occurrence information reflecting the spatial relationships between different stainings. When ImageNet21k/CLAM-based prediction was conducted using mIF WSIs, Staining set 1, which labeled $\alpha$-cells, $\delta$-cells, and neuronal axons, yielded significantly better classification results than Staining set 2. The best diagnostic performance was reached using the "channel-wise average" representation on Staining set 1 (Ensemble $AUROC = 0.956$; Fig. 2b and Supplementary Table 1).

### AI models attend to specific biological traits
Upon completion of model training, we aimed to understand the biological features utilized by the models in predicting T2D. For this purpose, we employed XAI techniques specifically within the domains

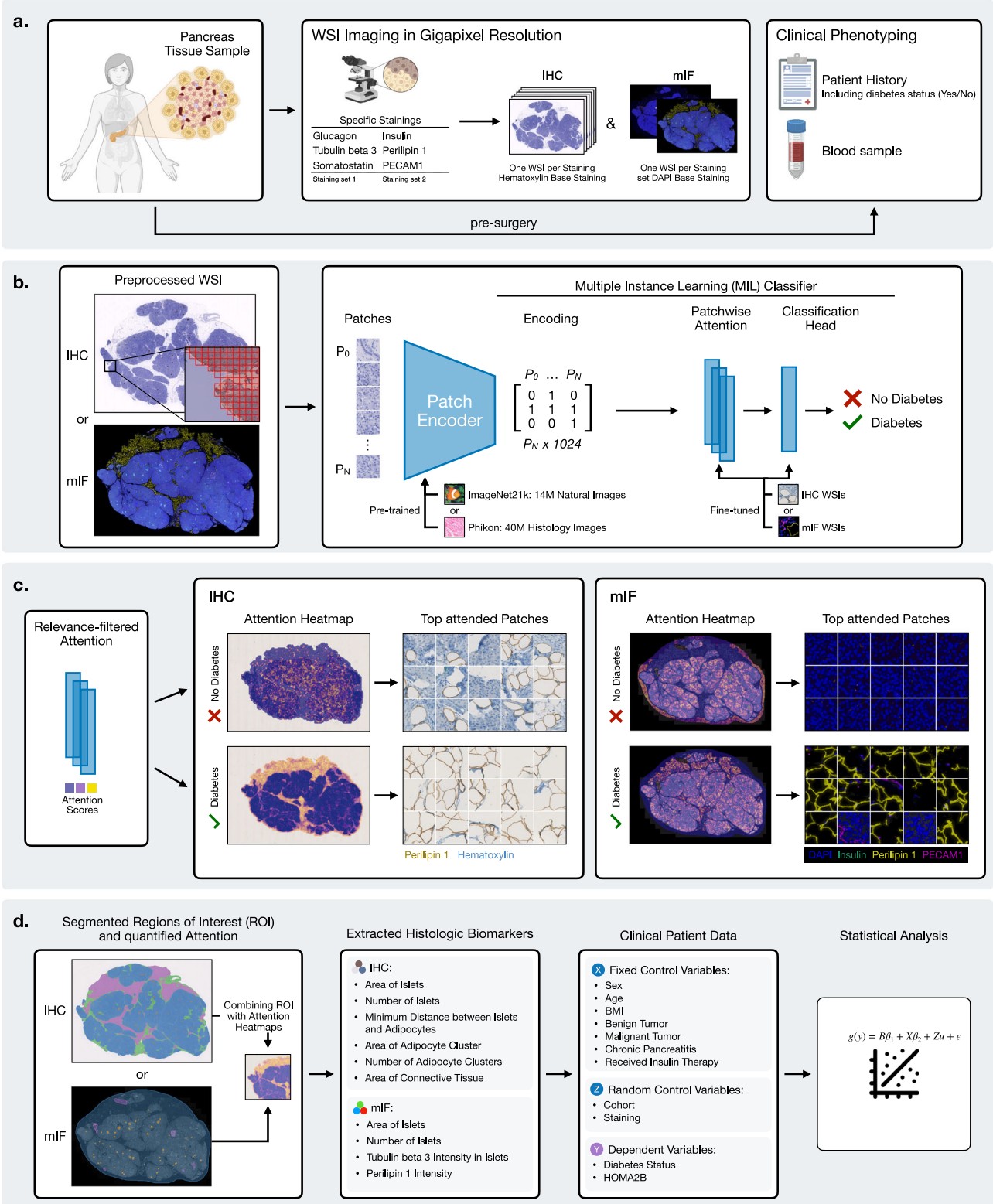

**Fig. 1 | Overview of the analysis procedure from sample and data acquisition to the final statistical analysis. a** Data from immunostained serial pancreatic tissue sections were integrated with fasting blood samples and clinical patient data as described in the methods. Briefly, antibodies against insulin, glucagon, somatostatin, PECAM1, perilipin 1, and tubulin beta 3 were used for single IHC immunostaining, whereas in the case of mIF, antibodies were combined into two distinct staining sets. Immunohistological whole slide images (WSIs) were generated via brightfield and fluorescence microscopy. **b** Flowchart showing the building process of the models able to predict the T2D status from the WSIs. **c** XAI methods were used to identify regions of interest (ROIs) utilized by the models for their prediction. **d** The identified ROIs were finally segmented and quantified. The extracted histologic biomarkers, together with covariates from clinical patient data were analyzed with statistical models to assess their association with T2D status and insulin secretion (HOMA2B). Created in BioRender. Wagner, R. (2026) https://BioRender.com/su53h8z.

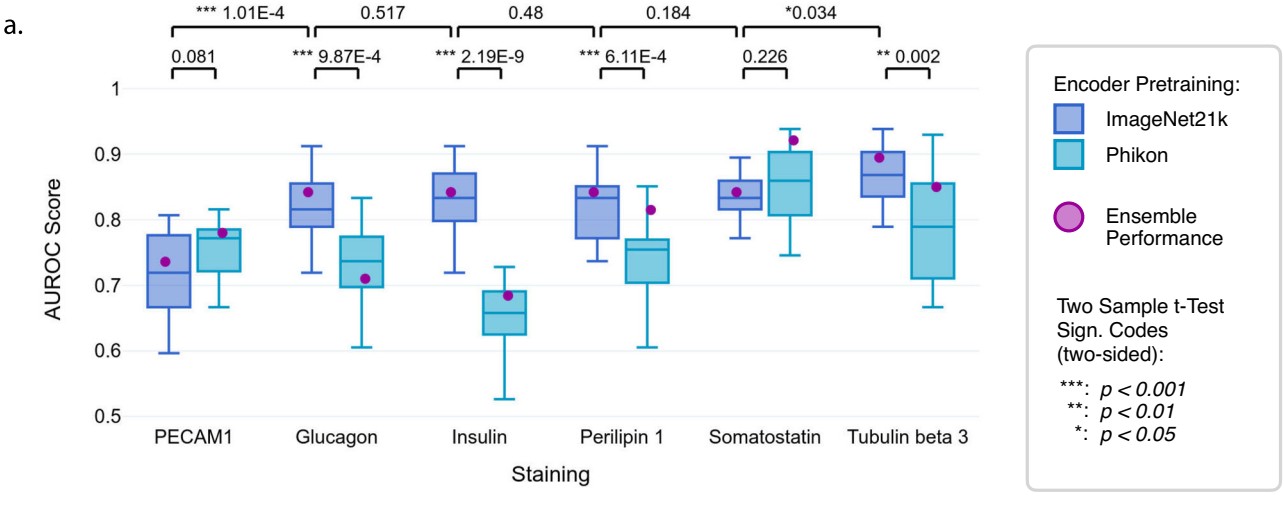

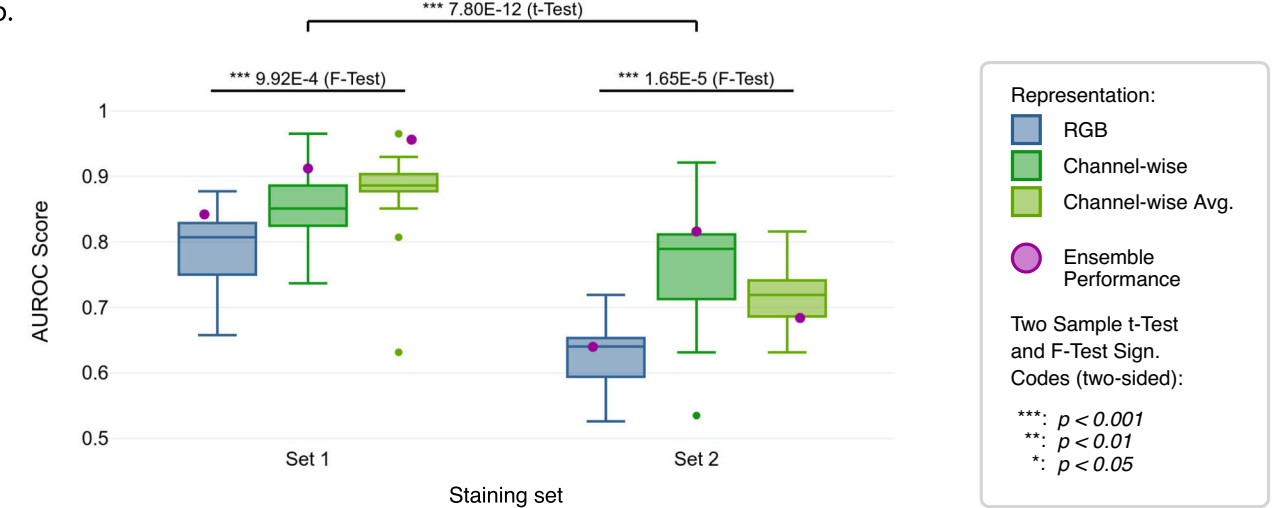

**Fig. 2 | Performance efficiency of the trained models to predict T2D on IHC and mIF WSIs.** Ensemble performance indicates the AUROC achieved when the respective 15 trained models per configuration are combined by aggregating their individual predictions by using the averaged class probabilities to obtain a final prediction. **a** AUROC by ImageNet21k/CLAM with the IHC WSIs stained for PECAM1, glucagon, insulin, perilipin 1, somatostatin and tubulin beta 3 compared to Phikon/CLAM. In all cases except PECAM1 and somatostatin ImageNet21k/CLAM yields better performance. Tubulin beta 3 yields the highest performance across stainings

($p < 0.05$), while performance differences between stainings are generally small. **b** AUROCs by ImageNet21k/CLAM on mIF WSIs stained with Staining set 1 and 2 and represented in RGB, channel-wise or channel-wise average, showing overall better performance ($p < 0.001$) for Staining set 1 and better performance ($p < 0.001$) for the channel-wise average representation in that set. The AUROCs in (**a**, **b**) are both computed on the held-out test sets (25 patients) across 15 training runs on different folds on the IHC or mIF data, respectively. The box plots show the quartiles, with the median (i.e., second quartile) marked by a line inside.

of attention[26] and attribution[27–29] methods, to highlight regions and features critical to the models when predicting the T2D status (Fig. 1c). Attribution methods in general determine how much each input feature attributes to a model's output (i.e., how important each feature is in a prediction) through e.g., gradients or perturbations. Attention methods, a subset of attribution methods, leverage the output of the attention module used e.g., in the transformer architecture[30]. Raw attention outputs indicate features of general importance to the model but lack a direct connection to specific outputs, requiring filtering (e.g., via gradients or relevance scoring) to highlight features used to predict a specific outcome. Since the regions highlighted by the XAI methods reflect the features deemed important by the model for the prediction task, we consistently applied these methods to the best-performing model for both mIF and IHC modalities. While these methods show what regions in the WSI are important, they do not show why they are important. To this end, we conducted a systematic

biological interpretation of the XAI results through structured analysis of feature attributions, biomarker determination and cross-validation with established biological literature.

Initially, we determined the significance of more global ROIs within the tissues before focusing on local features within these regions. To this end, we use the built-in attention mechanism of the MIL classifier on the specific patches, based on which we create heat-maps for the individual IHC and mIF WSIs of each patient (see exemplary Fig. 3). We structured the subsequent results according to both immunostaining techniques.

### Quantifying attention to regions-of-interest

To quantitatively assess the general importance of specific regions across all heatmaps, we implemented a scalable, data-driven methodology. We examined highly attended patches to determine their corresponding tissue structures. Those recurring were defined as

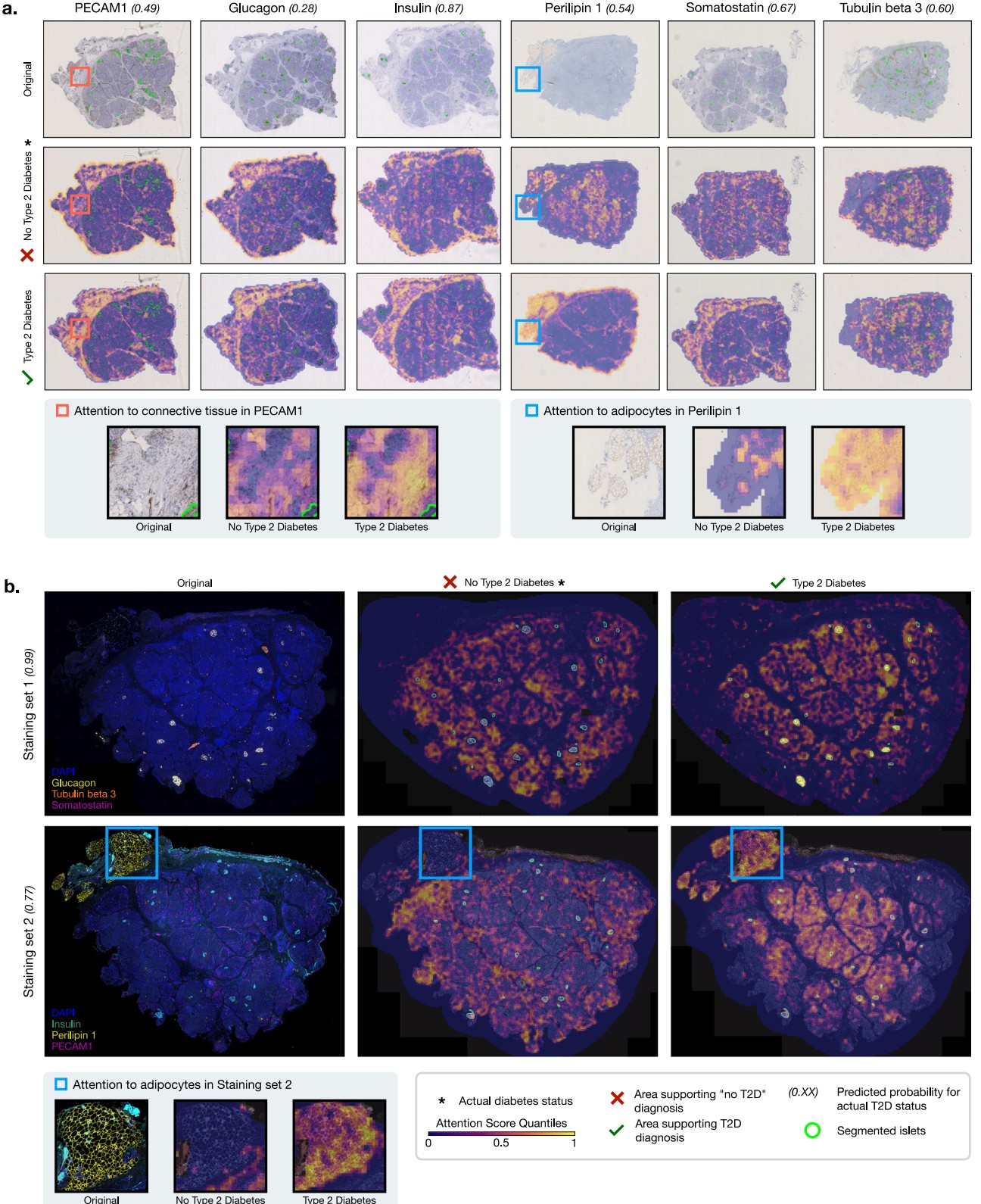

**Fig. 3 | Representative attention heatmaps for the IHC and mIF WSIs of a single patient.** Attention-based heatmaps on the global WSI level for **a** each IHC staining and **b** Staining sets 1 and 2 of mIF staining. See Supplementary Fig. 13 for the same overview of a T2D positive patient. Specific ROIs where the model attends to either connective tissue or adipocytes when classifying T2D are depicted below.

ROIs and segmented in the WSI using a specifically trained DL segmentation model to enable quantitative analysis of the attention to different relevant tissue features. The attention scores within ROIs were standardized for tissue size and total amount of attention.

However, results from IHC and mIF staining modalities should not be directly compared. The IHC models analyzed each staining independently, while mIF models evaluated all stainings within a staining set simultaneously.

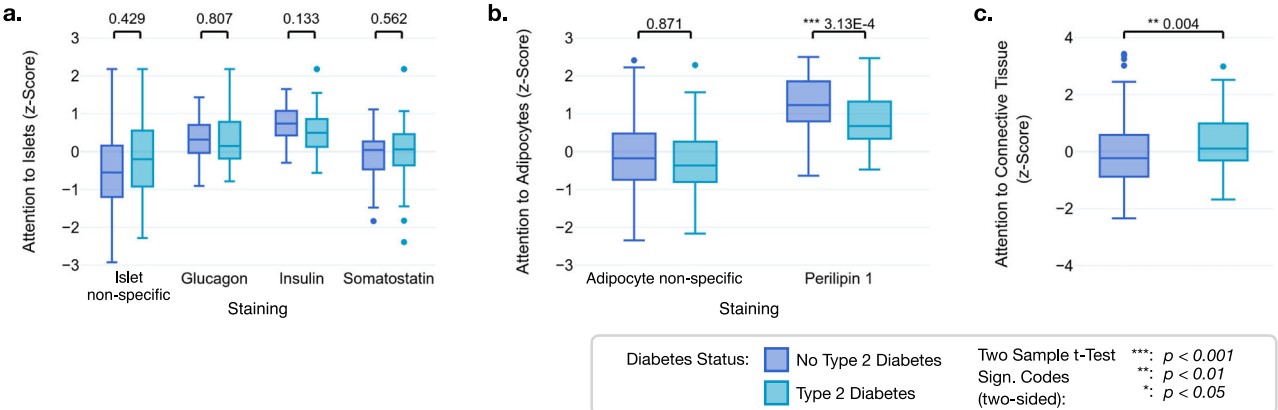

**Fig. 4 | Quantifying attention to regions of interest for IHC WSIs of non-diabetic and T2D donors.** Z-scores ($n = 100$) for average attributed attention to **a** islets, **b** adipocytes, and **c** connective tissue-rich areas for the individual IHC stainings of patients with and without T2D. Respective p-values indicate significant differences between T2D and non-diabetic status. Variables on non-interpretable Y-scales are z-standardized. The box plots show the quartiles, with the median (i.e., second quartile) marked by a line inside.

IHC and mIF stainings also differ in their dynamic ranges (1–2 vs. 3–5 order of magnitude, respectively), and thus in regard to their sensitivity and saturation of the signal. Moreover, the optimization of the mIF stainings required the use of antibodies that in several cases differed from those used for IHC detection of single antigens.

**IHC.** The analysis of attention scores in Fig. 4 is shown for each specific staining of the respective ROIs and summarized for the non-specific stainings as a more robust comparison baseline. It revealed a heterogeneous distribution of attention towards tissue components such as islets, adipocytes, or connective tissue in the individual IHC WSIs. Notably, islets exhibited the highest average attention scores for insulin- and glucagon-stained WSIs, irrespective of diabetes status (Fig. 4a). Attention to adipocytes was substantially elevated in perilipin 1-stained WSIs, which specifically highlight adipocytes, compared to other non-specific stainings (Fig. 4b). Furthermore, the attention to adipocytes was significantly higher in non-diabetic patients for perilipin 1-stained WSIs. In contrast, for connective tissue, which was never specifically stained, attention was significantly higher in T2D patients (Fig. 4c).

**mIF.** In the mIF WSIs, total attention to selected tissue, i.e., all patches containing stained tissue, differed between the individual stainings of the two staining sets, while the models attributed significantly higher attention to glucagon, tubulin beta 3, perilipin 1, and PECAM1 in the WSIs of T2D donors (Fig. 5a). As most of the higher attended signals also label extra-islet structures, we next determined the ratio between attention to islets (Fig. 5b) and total attention to selected tissue, which in the case of glucagon, insulin, and somatostatin stainings comprises only the patches containing islets. The ratio shows that, compared to the surrounding tissue, islets are attended above average for tubulin beta 3 and PECAM1 and below average for perilipin 1 staining (Fig. 5c). Interestingly, attention to perilipin 1 was significantly higher in T2D patients, contrasting with the findings from IHC analyses. This observation suggests a potentially critical role of adipocytes in T2D pathophysiology, warranting further investigation. Strikingly, in the Staining set 2 both the attention to islets and the islet-to-tissue attention ratio were highest for DAPI, the nuclear dye, suggesting that T2D pathology is associated with a global alteration of islet morphology (Fig. 5b, c).

**Unbiased identification and computation of histologic biomarkers**

While the global heatmaps facilitate the visualization of larger ROIs in the tissue, they do not provide detailed insights into the finer

biological features that the model considers significant. However, much more detailed visualizations are necessary to formulate specific hypotheses and subsequently biomarkers based on the features or regions the model attends to. The process of identifying biomarkers is largely unbiased, as the model, without prior knowledge, learns to classify patients solely from the images, potentially uncovering novel histological features beyond predefined hypotheses.

We first sampled the top attended regions using attention scores (see exemplary Supplementary Fig. 1), essentially zooming in from image-level to pixel-level. To further determine the specific pixel-level features utilized by the model in each of the top-attended regions, we employed multiple attribution methods to quantify the contribution of individual pixels to the predicted outcomes, as the attention modules only operate on the patch level. The heatmaps confirm that the model was indeed able to recognize biological traits, such as nuclei, adipocyte cytosolic compartments, inter-cellular structures, or islets (Supplementary Fig. 2).

Using the global and local heatmaps, we selected histologic biomarkers potentially representing key characteristics of the features attended to by the model in distinguishing between T2D and non-T2D patients. Figure 6 provides selected regions from IHC and mIF WSIs with histologic biomarkers and the corresponding pixel-level attribution heatmaps derived from these analyses. Specifically, Fig. 6a, d reveal consistently high attention to the stained islets across both immunostaining techniques, suggesting that the islets possess features essential to T2D pathology, besides their number and their average size. Of note, Fig. 6d underscores the relevance of neuronal-axonal structures, stained by tubulin beta 3, often attended especially within the islets. Furthermore, Fig. 6b highlights the significance of adipocytes, whereas Fig. 6c emphasizes the importance of connective or fibrotic tissue. Based on these ROIs to the DL model, we defined and computed several histologic biomarkers to validate why these regions are important to the model.

**Exploratory analysis of the histologic biomarkers.** The defined histologic biomarkers for the IHC and mIF modalities were computed through the quantitative XAI results and the segmentation maps of the ROIs (Fig. 1d). Notably, these biomarkers exhibited variations between both modalities as certain ROIs could not be computed for mIF WSIs, e.g., connective tissue due to the absence of a dedicated staining. All histologic biomarkers were standardized for tissue size.

A first explorative analysis of the histologic biomarkers across IHC stainings revealed that persons without diabetes generally exhibited larger islets (Fig. 7a). In the case of perilipin 1, the segmentation of the

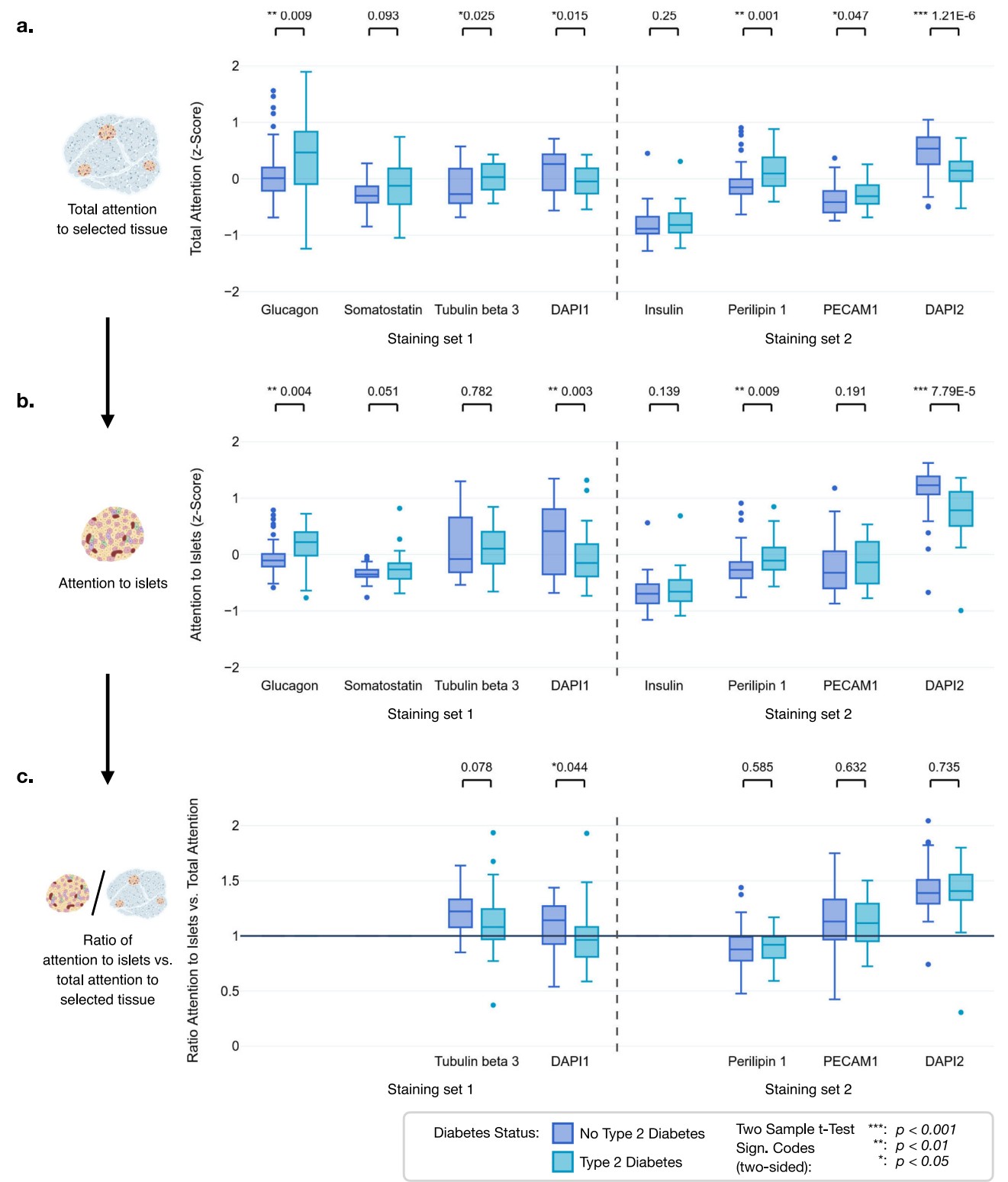

**Fig. 5 | Comparison of attention within the total area containing stained tissue and islets between the individual mIF stainings. a** *Z*-scores (*n* = 100) for the attributed attention of the channel-wise models to the different staining channels in the mIF WSIs labeled with Staining sets 1 and 2. **b** Attention of the channel-wise models to islets in the mIF WSIs. **c** Ratio of attention to islets and the total attention to the total area containing stained tissue. Values greater than 1 suggest that, for this staining, the islets are more important to the model than the average tissue patch. The ratio is always 1 for stainings that exclusively stain the islets, i.e., glucagon, somatostatin, and insulin. Respective p-values indicate significant differences between T2D and non-diabetes status. Variables on non-interpretable *Y*-scales are z-standardized. The box plots show the quartiles, with the median (i.e., second quartile) marked by a line inside. Created in BioRender. Wagner, R. (2026) https://BioRender.com/su53h8z.

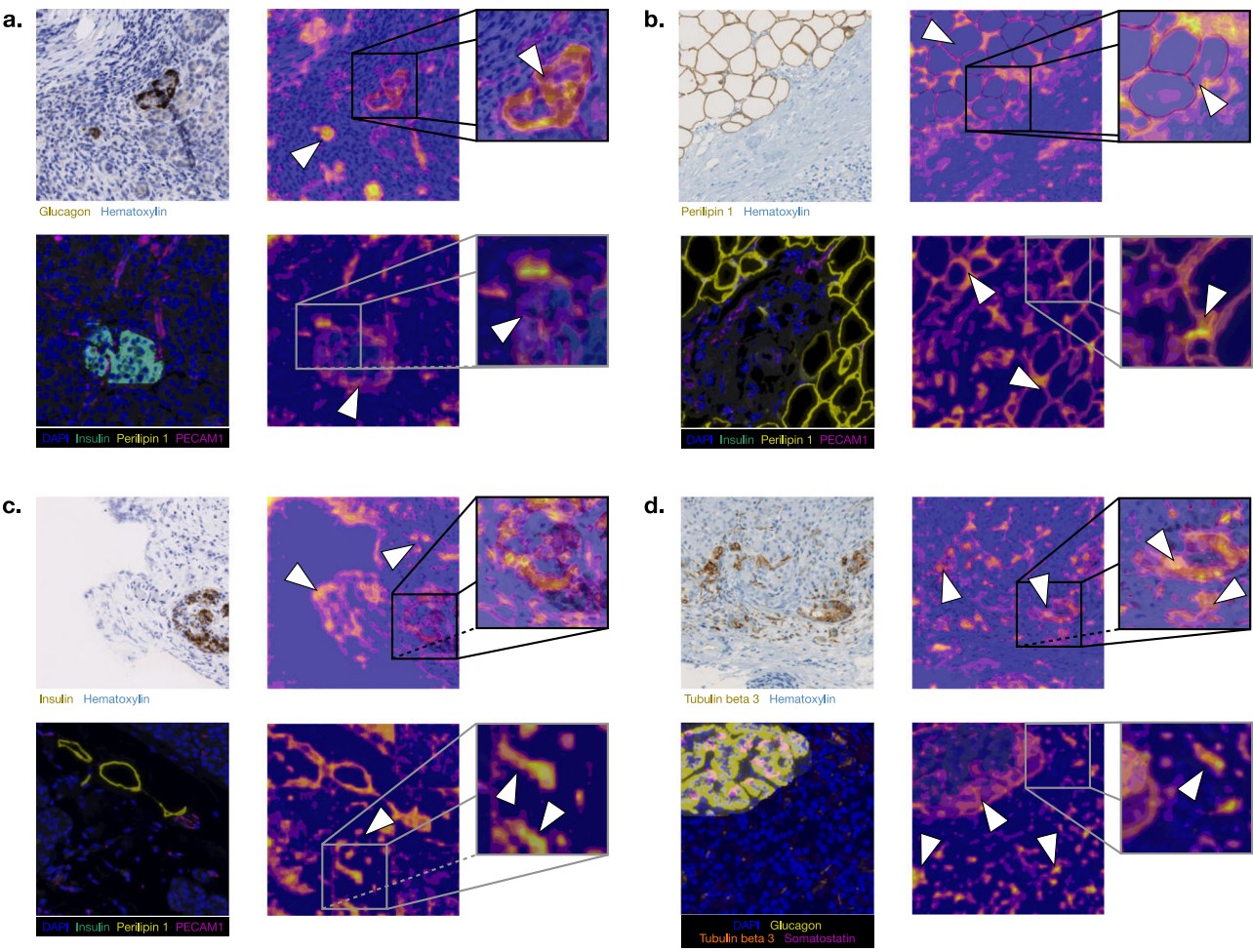

**Fig. 6 | Pixel-level heatmaps showing importance of features related to specific histologic biomarkers.** Heatmaps for both staining techniques highlight the importance of **a** islets, **b** adipocyte clusters, **c** connective tissues, and **d** neuronal-axonal structures.

islets was more challenging due to a lack of islet-specific staining, resulting in considerably fewer detected islets. Further, patients with T2D tended to have larger adipocyte clusters (Fig. 7b) and connective tissue-enriched areas (Fig. 7c). An adipocyte cluster was defined as either an accumulation of adipocytes or a single adipocyte in the tissue. When assessing the average minimum distance between each islet and the nearest adipocyte, computed only on the insulin staining segmentations, we observed that in patients with T2D, the islets were significantly closer to adipocytes than in patients without T2D (Fig. 7d).

### Statistical analysis reveals a significant association of histologic biomarkers with diabetes status and insulin secretion

To statistically assess the impact of the histologic biomarkers on diabetes status and insulin secretion, measured through HOMA2B (c-peptide), we considered several control variables, which were either directly pertaining to individual donor traits, including sex, age, body mass index (BMI), underlying pancreatic tumor, chronic pancreatitis, and insulin therapy; or experimental conditions, including cohort, and tissue staining (Fig. 1d). We analyzed the data using generalized mixed linear models (GMLMs), accounting for random effects linked to the cohort and staining method.

**IHC.** For the IHC stainings, regression analysis revealed a significant positive association of both adipocyte cluster areas (0.266; $p = 0.035$) and connective tissue-rich areas (0.524; $p < 0.001$) with diabetes status (Table 1). We observed an inverse association of diabetes status with the number of adipocyte clusters ($-0.401$; $p = 0.011$). The distance

between islets and adipocytes also showed an inverse association ($-0.633$; $p = 0.007$), suggesting that a smaller distance between adipocytes and islets is associated with T2D. For the fixed effects control variables, sex (males vs females) (0.610; $p = 0.007$), age (0.741; $p < 0.001$), BMI (0.538; $p < 0.001$), and chronic pancreatitis (1.420; $p < 0.001$) had a significant association with diabetes status.

When regressing on insulin secretion, we observed a significant negative association with the area of connective tissue ($-0.132$; $p = 0.001$) and the number of adipocyte clusters ($-0.189$; $p < 0.001$) (Table 2). Contrarily, there was a significant positive association of insulin secretion with the number of islets (0.069; $p = 0.043$) and the area of adipocyte clusters (0.216; $p < 0.001$). This association turns negative when interacting with BMI, suggesting an attenuation of this association in the case of higher BMI ($-0.071$; $p = 0.020$). Among the control variables, sex ($-0.179$; $p = 0.008$), age ($-0.125$; $p = 0.001$), BMI (0.218; $p < 0.001$), insulin therapy ($-1.460$; $p < 0.001$), and chronic pancreatitis (0.254; $p = 0.041$) had a significant association with insulin secretion. Among the random effects in both models, there were large variances between cohort intercepts and virtually no variances across different stainings.

As one would expect potential effects on histological characteristics of pancreas tissue due to increased disease duration, we further regressed the histologic biomarkers on the duration of T2D. However, we also controlled for the diabetes status, as diabetes duration is a near-perfect predictor of diabetes status, except in cases where T2D was diagnosed for the first time from the tissue sample. We observed significant negative correlations for diabetes duration with the area of

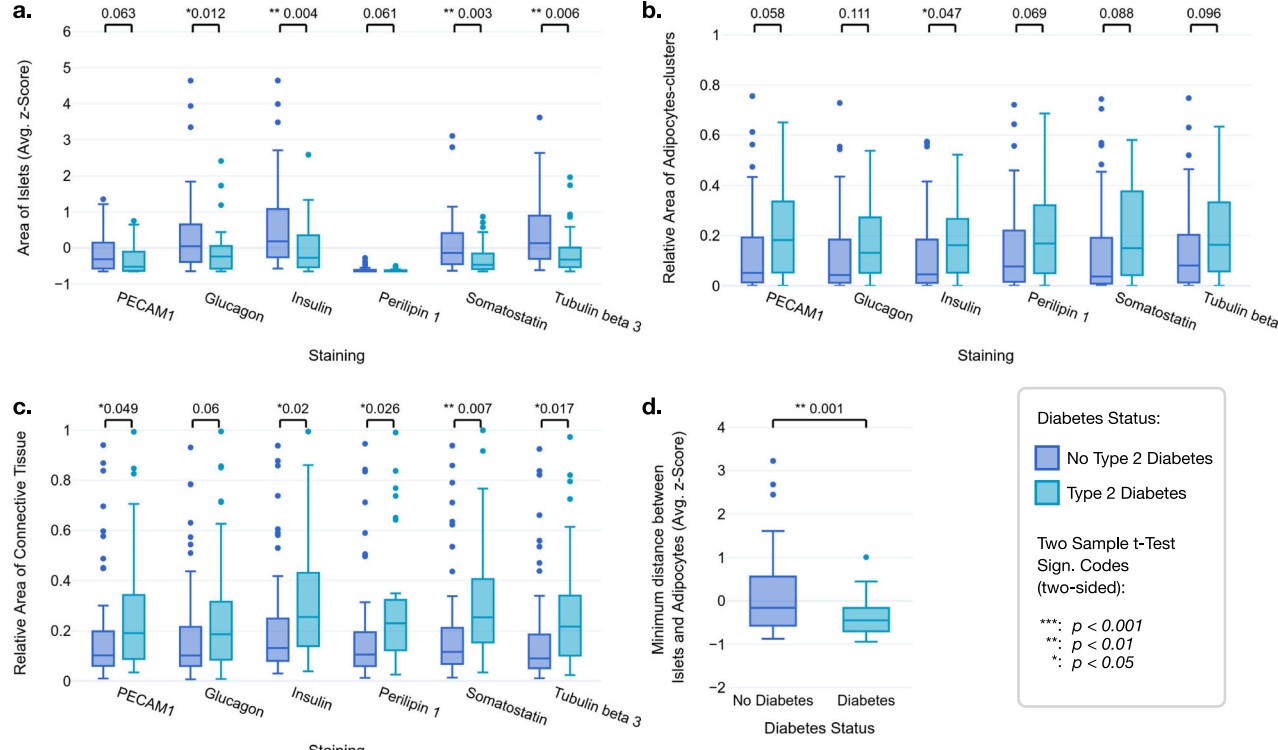

**Fig. 7 | Quantification of XAI-derived histologic biomarkers for IHC WSIs of non-diabetic and T2D donors. a** Average islet area, and relative areas of **b** pancreatic adipocyte clusters (a cluster was defined as connected adipocyte cells, single non-connected adipocyte cells were counted as one cluster), and **c** connective tissue in the IHC WSIs of patients with and without T2D. **d** The average minimum distance between each islet and the nearest adipocyte in the insulin-stained WSIs from non-diabetic and T2D donors. Respective $p$ values indicate significant differences between T2D and non-diabetic status in each plot ($n = 100$ for (**a–d**)). The box plots show the quartiles, with the median (i.e., second quartile) marked by a line inside.

## Table 1 | GMLM regression for diabetes status (IHC)

| Fixed Effects | Estimate | SE | P Value (p) | Sign. |
|---|---|---|---|---|
| Intercept | −1.607 | 0.290 | 3.16E-08 | *** |
| Area of islets | −0.237 | 0.192 | 0.216 | |
| Number of islets | 0.088 | 0.105 | 0.404 | |
| Area of adipocytes clusters | 0.266 | 0.126 | 0.035 | * |
| Num. of adipocytes clusters | −0.401 | 0.158 | 0.011 | * |
| Min. dist. islets and adipocytes | −0.633 | 0.237 | 0.007 | ** |
| Area of connective tissue | 0.524 | 0.124 | 2.42E-05 | *** |
| Sex | 0.610 | 0.226 | 0.007 | ** |
| Age | 0.741 | 0.137 | 6.38E-08 | *** |
| BMI | 0.538 | 0.114 | 2.25E-06 | *** |
| Malignant tumor | 0.330 | 0.240 | 0.170 | |
| Chronic pancreatitis | 1.420 | 0.394 | 3.10E-04 | *** |
| **Random effects** | **Type** | **Variance** | **STD** | |
| Staining | Intercept | 0 | 0 | |
| Cohort | Intercept | 0.176 | 0.483 | |
| Observations: 600 | | | | |
| Groups: {Staining: 6, Cohort: 2} | | | | |
| Log-likelihood: −298.010 | | | | |
| AIC: 624.010 | | | | |

Z-Test (two-sided) Sign. Codes: *$p < 0.05$, **$p < 0.01$, ***$p < 0.001$.
Regression results of the generalized mixed linear model analysis for diabetes status based on IHC stainings. Across 100 patients with six IHC WSIs per patient ($n_{total} = 600$). All continuous variables were z-standardized to ensure comparability across measures.

islets (−0.1741; $p < 0.001$) and the number of islets (−0.1951; $p = 0.012$), aligning with the general progression of T2D and further validating the biomarkers. Notably, we found a significant positive correlation for diabetes duration with area of adipocyte clusters (0.2397; $p = 0.009$; Supplem. Fig 12a).

**mIF.** Results from mIF WSIs are only partially comparable to those from IHC data. One reason is the inability to detect connective tissue-rich structures in the absence of specific mIF staining. In the case of the IHC stained WSI, we compute each biomarker for every different staining, resulting in six sets of imaging markers per patient. However, the two multiplexed WSIs resulted in only one set of imaging markers per patient, as the biomarkers of the two staining sets were distinct except for the area or size of the islets, which was computed over both sets, reducing the sensitivity of the mIF statistical models. When regressing the mIF biomarkers on diabetes status, only the control variables age (0.832; $p = 0.012$), BMI (0.593; $p = 0.024$) and chronic pancreatitis (1.915; $p = 0.048$) were significant (Supplementary Table 3a). However, when analyzing insulin secretion, we observed a significant association with the area of islets (−0.207; $p = 0.030$) and the number of islets (0.280; $p = 0.008$), as well as the control variables BMI (0.243; $p = 0.008$) and insulin therapy (−1.435; $p < 0.001$) (Supplementary Table 3b). We refer to Supplementary Figs. 10 and 11 for the evaluation of all GMLMs.

## Discussion

Our study aimed to obtain biological insights into T2D-associated changes in human pancreas tissues combining diverse immunostainings and AI techniques. To this end, we trained extensive DL models to differentiate between the presence or absence of T2D from gigapixel-sized histopathologic WSIs of pancreatic tissues from living donors

**Table 2 | MLM regression for HOMA2B (IHC)**

| Fixed Effects | Estimate | SE | P Value (p) | Sign. |
|---|---|---|---|---|
| Intercept | 0.342 | 0.230 | 0.262 | |
| Area of islets | 0.028 | 0.037 | 0.442 | |
| Number of islets | 0.069 | 0.034 | 0.043 | * |
| Area of adipocytes clusters | 0.216 | 0.042 | 2.62E-07 | *** |
| Num. of adipocytes clusters | − 0.189 | 0.40 | 3.10E-06 | *** |
| Min. dist. islets and adipocytes | 0.053 | 0.040 | 0.184 | |
| Area of connective tissue | − 0.132 | 0.038 | 5.26E-04 | *** |
| Area of adipocytes clusters:BMI | − 0.071 | 0.031 | 0.020 | * |
| Sex | − 0.179 | 0.067 | 0.008 | ** |
| Age | − 0.125 | 0.036 | 6.55E-04 | *** |
| BMI | 0.218 | 0.039 | 3.16E-08 | *** |
| Insulin therapy | − 1.460 | 0.098 | 3.17E-42 | *** |
| Malignant tumor | 0.074 | 0.075 | 0.320 | |
| Chronic pancreatitis | 0.254 | 0.124 | 0.041 | * |
| **Random Effects** | **Type** | **Variance** | **STD** | |
| Staining | Intercept | 0 | 0 | |
| Cohort | Intercept | 0.096 | 0.311 | |
| Observations: 546 | | | | |
| Groups: {Staining: 6, Cohort: 2} | | | | |
| Log-likelihood: −615.653 | | | | |
| AIC: 1265.307 | | | | |

Z-Test (two-sided) Sign. Codes: *$p < 0.05$, **$p < 0.01$, ***$p < 0.001$.

Regression results of the generalized mixed linear model analysis for HOMA2B levels based on IHC stainings. Across 100 patients with six IHC WSIs per patient ($n_{total} = 600$). All continuous variables were z-standardized to ensure comparability across measures.

recruited at two academic centers. This process involved multiple immunohistological stainings combined with two different microscopy techniques. We subsequently employed XAI methods to find AI-attended ROIs from these WSIs, from which we computed and analyzed histologic biomarkers using exploratory visualization and GMLMs.

The AI-based T2D prediction models demonstrated a highly reliable performance, depending on the staining approach. The highest predictive performance was obtained with mIF staining using our predefined Staining set 1, comprising stainings for $\alpha$- and $\delta$-cells, i.e., for glucagon and somatostatin, respectively, as well as staining of tubulin beta 3 for neuronal-axonal structures. The predictive power of this staining set was surprisingly better than that of Staining set 2 which specifically marked insulin, i.e., $\beta$-cells, in addition to adipocyte lipid droplet membranes and endothelial cells via perilipin 1 and PECAM1 staining, respectively. We also saw differences between the techniques aggregating information from multiplex stainings. While the "channel-wise" approach did not retain co-occurrence information of the different stainings, the "channel-wise average" approach reintroduced this information by using mean encodings for each patch location of the WSI at the trade-off of losing channel-specific details. The clear benefit of the "channel-wise average" approach in the case of Staining set 1 likely resulted from its provision of more robust information on islets compared to Staining set 2 due to the co-occurrence of two different channels (glucagon, somatostatin), highlighting islets only, as well as tubulin beta 3. Of note, tubulin beta 3 is not exclusively restricted to neuronal axons, but is also present, albeit at a much lower level, in islet cells, including $\beta$-cells (Suppl. Fig. 8)[31]. Thus, Staining set 1 may also have indirectly incorporated information on non-$\alpha$, non-$\delta$ islet cells, including $\beta$-cells, and thereby, identify subtle changes in the

relationship among these cells and in the islet structure. On the other hand, Staining set 2 had an inferior prediction performance with the channel-wise averaging approach compared to the channel-wise approach because it employed three stainings mostly occurring in non-overlapping areas, i.e., $\beta$-cells, adipocytes, and endothelial cells and thus, its benefit from co-occurrence information was lower.

While the IHC stainings had lower predictive performances compared to the mIF images, their highest average predictive performance was found for tubulin beta 3, which was also a component of the best-performing mIF Staining set 1. When comparing the average attention attributed to each of the mIF stainings between samples without and with T2D, we saw higher attention to tubulin beta 3 in T2D compared to non-diabetic WSIs (Fig. 5a). The data suggest that alterations in tubulin beta 3 stained structures may represent a distinguishing feature of T2D. Indeed, islet innervation impacts on islet cell function[32,33]. Through the autonomic nervous system, brain-derived signals reach pancreatic islet cells, including $\beta$-cell primary cilia[34] and modulate insulin secretion in humans[35]. Structural alterations of primary beta cell cilia have been reported in both humans[36] as well as in mouse models of the disease[37]. T2D islets have been shown to display an increased number of noradrenergic fibers, a trait negatively correlated with $\beta$-cell differentiation, insulin secretion, and islet size[32,38]. Additionally, enlarged peri-lobular ganglia have been observed near pancreatic adipocytes, with adipocyte infiltration being linked to an increased number of projections from intra-parenchymal ganglia in T2D[39]. Given that adipocyte function relies on neuronal signals, i.e., beta-adrenergic-stimulated lipolysis, remodeling of the neuronal landscape near the pancreatic adipocytes in T2D may modify the adipocyte secretome and thereby influence the pancreatic microenvironment and the islets. Our data suggest that alterations at the level of (islet) innervation might be involved in T2D. However, it remains to be determined whether these changes directly affect hormone secretion, islet cell differentiation, and ultimately $\beta$-cell mass.

The islets received variable levels of attention within the individual IHC and mIF stainings. While in the IHC analysis islets stained for either insulin or glucagon received the greatest attention (Fig. 4a), in the case of the mIF stainings, the glucagon signal received the highest attention relative to insulin, somatostatin and tubulin beta 3 (Fig. 5b). The ratio between the attention to the islets (Fig. 5b) and the total attention to the selected tissue patches (Fig. 5a), which also includes exocrine tissue, indicates that for the non-islet restricted markers tubulin beta 3 and PECAM1, the islets were attended above average (Fig. 5c), suggesting that islet innervation and vascularization have higher relevance for T2D classification compared to their extra-islets counterparts, and align with the notion that the islets are primarily altered in T2D relative to the surrounding pancreatic tissue. For PECAM1, this likely reflects changes in vessel cytoarchitecture, but for tubulin beta 3, it remains unclear whether the observed alterations stem from neuronal fibers, islet cells, or both.

The analysis of the IHC stainings for insulin, glucagon, somatostatin, and tubulin beta 3 revealed significantly smaller islets in patients with T2D compared to non-diabetic individuals (Fig. 7a). Although we cannot say whether the reduction of $\beta$- or $\alpha$-cell mass accounts for the smaller islets, previous observations suggested that larger islets are preferentially lost in T2D[40]. Intriguingly, some evidence indicates that smaller islets have better glucose-responsive insulin secretion compared to large islets[41–43]. This is in line with our finding that lower insulin secretion was associated with larger mIF image-derived islet size when also adjusting for confounders such as number of islets, age, BMI, underlying disease, and exogenous insulin therapy (Supplementary Table 3b).

The significant differences between T2D and non-diabetic WSI consistently observed across all attention and attribution heatmaps derived from perilipin 1-stained WSIs, combined with the statistical analysis of factors contributing to T2D highlight intra-pancreatic

adipocytes as key tissue compartments related to the disease. This is further underscored by the lowest attention to islets (islets vs whole tissue ration) on the mIF label perilipin 1 (Fig. 5c), suggesting that adipocyte-specific traits are essential for accurate T2D classification.

In our study, we used two histologic biomarkers of intra-pancreatic steatosis: the area of adipocyte clusters and the number of adipocyte clusters. The moderate correlation coefficient (0.4) between these biomarkers suggests that they reflect different aspects of intra-pancreatic steatosis (Supplementary Fig. 12). Interestingly, only the area of adipocyte clusters showed a positive significant correlation with the diabetes duration, underscoring the extent of pancreatic fat accumulation as a relevant contributor to disease progression, in line with recent clinical evidence highlighting the role of pancreatic fat in T2D[44]. Further analysis of the IHC-derived adipocyte biomarkers revealed differential associations with T2D. Specifically, the area of adipocyte clusters was positively associated with T2D (Table 1, and Fig. 7b), while the number of adipocyte clusters was inversely correlated with the condition (Table 1). The area of adipocyte clusters was well correlated with BMI, whereas there was no substantial correlation between the number of adipocytes and BMI (Supplementary Fig. 12).

For insulin secretion, we observed contrasting relationships: a larger area of adipocyte clusters was linked to higher insulin secretion, while a higher number of adipocyte clusters was associated with lower insulin secretion (Table 2). The positive association of adipocyte cluster area with insulin secretion is also in accordance with clinical observations reporting a positive correlation of pancreatic steatosis with insulin secretion in persons with a low genetic risk of T2D[45]. These results suggest that the morphologic features of pancreatic fat, i.e., size vs number of cell clusters, besides its mere existence differentially impact islet function. A higher number of adipocyte clusters may indicate a more advanced adipose infiltration of pancreatic tissue, i.e., intralobular adipocytes[46]. Such an intralobular infiltration brings the adipocytes closer to the islets, thereby altering the islet micro-environment, and potentially impairing insulin secretion (Supplementary Fig. 3). Indeed, we observed that in patients with T2D, pancreatic islets are on average closer to adipocytes than in patients without T2D, and the distance between islets and adipocytes is negatively correlated with the T2D status (Fig. 7d and Table 1). These observations support the hypothesis that paracrine signaling from the adipocyte secretome could negatively affect the islets and contribute to T2D[22]. Indeed, a recent publication found a higher pancreatic adipocyte content accompanied by increased beta cell dedifferentiation in individuals with diabetes[47].

We have recently shown that pancreatic adipocytes express gastric inhibitory peptide receptor (GIPR) and display beta-adrenergic-/incretin-sensitive lipolysis and immune response[48], functions that become dysregulated in T2D and may have detrimental effects on neighboring islets.

The association of pancreatic steatosis with T2D risk is also substantially modulated by factors such as genetic risk or prevailing metabolic background[45,49,50]. Our results also unveiled an interaction between the area of adipocyte clusters and BMI, suggesting a diminishing effect of intra-pancreatic steatosis on insulin secretion in persons with obesity (Table 2). This effect modification of the BMI on intra-pancreatic steatosis is consistent with epidemiologic data from a Japanese cohort showing that pancreatic fat robustly predicts diabetes only in lean individuals[51].

Although we did not use a specific collagen staining, the peroxidase reaction, employed for antigen visualization in the IHC approach, enabled co-visualization of the surrounding reference space, including fibers in ECM (extracellular matrix)-enriched areas within the pancreas tissue, as previously reported for other tissue types[52]. Combined with hematoxylin staining detecting cell nuclei, this led to a clear distinction between the lobular acinar tissue and the intercalated connective tissue. The DL models showed greater attention to the fibrotic-like, connective tissue-enriched areas when predicting T2D versus the non-diabetic state (Fig. 4c). Furthermore, our histologic biomarkers analysis revealed more abundant connective tissue-enriched areas in WSIs of T2D (Fig. 7c), in accordance with previous findings. Fibrosis has indeed been recognized as a structural alteration in the pancreas of persons with T2D[9,10,46,53] and ECM-related genes are upregulated in islets of living donors with T2D[54].

We also found a positive association of fibrotic area with T2D and lower insulin secretion as measured by HOMA2B (Table 1). Of note, these associations were independent of underlying pancreatic disease, whether benign or malignant, chronic pancreatitis, and insulin therapy. According to our results, pancreatic tissue of patients with T2D undergoes morphological alterations that extend beyond islets, such as enrichment in fibrotic-like structures and more islet-proximal adipocytes, which most likely have negative consequences for the islet microenvironment and islet function. These observations are especially intriguing as impaired glucose tolerance and T2D are also associated with a persistent activation of platelets[55,56], which promotes fibrosis[57].

While this data-driven approach has the potential to accelerate and facilitate the understanding of T2D pathogenesis, several potential limitations must be considered. One is that immunostaining approaches can be associated with technical pitfalls such as antibody binding specificity, nonspecific background staining, or variations in the staining intensity across samples, regardless of target antigen level. Furthermore, our histological preparations were not representative of the whole organ, since they were obtained from limited regions of the pancreas, the anatomical location of which varied among the donors. Further, despite originating from the "healthy" margins of the surgical resection, as also verified by postoperative pathological assessment, the samples could be affected by changes in the tissue microenvironment, such as inflammation, fibrosis, and angiopathy[58].

In our study, we analyzed a unique WSI dataset of the human pancreas, derived from two independent cohort sites, encompassing pre-surgical metabolic phenotyping of living donors both with and without T2D. Our staining approach covered a wide spectrum of pancreatic morphology, including islets, innervation, adipocytes, and vascularization, captured with brightfield and multiplex-fluorescence microscopy. Based on this unique dataset, we trained several DL models that, unlike a routine pathological assessment, reliably allow the prediction of the T2D status. This reliable prediction is novel and represents a remarkable advancement in the field of DL-based histologic biomarker discovery, especially because the pancreas, although a central organ in the pathophysiology of T2D, only exhibits subtle anatomical changes, unlike in the case of cancer or pancreatitis. Although $\beta$-cells are considered to play a key role in T2D, $\alpha$- and $\delta$-cells have gained increased attention during the last years, being essential intra-islet modulators of $\beta$-cell function[59,60]. Remarkably, the most accurate prediction was obtained with a staining set that focused on islet $\alpha$- and $\delta$-cells and neuronal axons, without specific attention to $\beta$-cells. Such an unexpected finding calls for the thorough 3D ultrastructural resolution of all islet cell types, as carried out for isolated mouse beta cells[61], in pancreatic tissue from donors with normoglycemia or T2D. While our quantitative analysis reinforces existing hypotheses about the involvement of pancreatic islets and enrichment of connective tissue in T2D, we propose that pancreatic adipocytes, together with pancreatic innervation, play a central role in islet dysfunction and T2D. Despite substantial experimental evidence demonstrating the influence of innervation on both islet[33] and adipocyte function[62], the precise molecular mechanism underlying the crosstalk between islets and pancreatic adipocytes, and how this interaction is disturbed in T2D, remains poorly understood.

The findings highlight the complexity of T2D pathology and shall motivate further deployment of XAI methods on other cohorts to aid the discovery of novel structural histologic biomarkers. Beyond the

specific histologic traits identified in our work, this study also provides a proof-of-concept on how XAI can be used in research on T2D to analyze high-dimensional and complex data. This novel approach could sharpen and guide our focus in the research of diabetes prevention and treatment toward the most promising targets.

## Methods

### Clinical cohort

We analyzed clinical, laboratory, and histologic data from living donors obtained within the "LIDOPACO" (LIving DOnor PAncreatic COhort) programs at two academic sites of the German Center for Diabetes Research network, the University Hospital Tübingen and the University Hospital Dresden (Supplementary Figs. 4 and 5). Patients undergoing pancreatic surgery for different indications provided written informed consent to donate blood samples and pancreas tissue, and share health records and laboratory data for research purposes at both study sites (Tübingen and Dresden). The study was approved by the Ethical Committees of the Technische Universität Dresden (Reference EK 151062008) and Eberhard Karls Universität Tübingen (Reference 697/2011BO1). We obtained macroscopically healthy tissue resected during surgery, but not required for further pathology workup. All patients were of European ethnicity. Additionally, fasting blood was drawn pre-surgery for detailed metabolic phenotyping. Fasting glucose and C-peptide levels were measured as previously described[63], and homeostatic model assessment (HOMA) of insulin secretion was calculated using the computer model-based HOMA-2B using glucose and C-peptide[64]. None of the participants had depleted endogenous insulin production as measured by C-peptide-based HOMA2B (lowest HOMA2B: 6 with a diabetes duration of 24 years), excluding type 1 diabetes among the participants. Information on medical history was collected by a physician. Documented by their health records, T2D patients were diagnosed as having T2D at least one year before admission to pancreatic surgery. This excludes diabetes in the context of exocrine pancreatic disease. In contrast, patients without diabetes neither had diabetes nor did they fulfill diagnostic criteria of T2D based on glycated hemoglobin (HbA1c) and fasting glucose, as defined by the American Diabetes Association[65].

### Data acquisition

**Hospital patient data.** The patient cohort-related metadata are summarized in Supplementary Fig. 6 and Supplementary Table 2. The quantitative analysis of these parameters in Supplementary Fig. 6 revealed an unbalanced distribution of sex, age, BMI, tumor type, chronic pancreatitis, and cohort between the patients with and without diabetes. In 9 patients, fasting blood samples were not obtained prior to surgery, and therefore, HOMA2B was not calculated. Sex was indicated as self-reported sex and was not considered as a factor at study design, and we did not perform sex-stratified analyses due to the limited sample size.

**IHC immunostaining and brightfield microscopy.** Formalin-fixed, paraffin-embedded (FFPE) pancreatic sections (2–4 μm thick) were processed using an automatic slide stainer BenchmarkUltra (Ventana Technology, Roche Diagnostics). Deparaffinization was performed for four min at 72 °C using EZPrep (Roche Ventana, #5279771001), followed by antigen-retrieval (AR) for 40 min at 100 °C with TRIS-based CC1-buffer (Roche Ventana, #5424569001). After peroxidase-inhibition with I-View Inhibitor (Roche, #06396500001), the sections were incubated with primary antibodies against insulin (1:1000; Dako, #A0564), glucagon (1:600; Santa Cruz, #sc13091), somatostatin (1:6000; Invitrogen, #14-9751-80), CD31 (1:100; Dako, #M0823), perilipin 1 (1:2000; Progen, #651156), and tubulin beta 3 (1:2500; R&D Systems, #MAB1195). The secondary horseradish peroxidase-linked antibody was detected via an Opti-View DAB IHC detection kit (Roche Ventana, #06396500001). The samples were counterstained with

hematoxylin. WSI acquisition was performed with a Hamamatsu NanoZoomer 2.0-HT using 20x magnification and NDP.scan 2.5 software. Exemplary zoom-ins of WSIs are shown in Supplementary Fig. 7.

**mIF staining and fluorescence microscopy.** Consecutive 2–4 μm FFPE sections derived from the identical pancreatic specimens used for the IHC stainings underwent fluorescent staining via an automated system DiscoveryUltra (Ventana Technology, Roche Diagnostics). Deparaffinization utilized EZPrep (Roche Ventana; #5279771001) for 32 min at 69 °C, and subsequent AR was performed with TRIS-based CC1-buffer (Roche Ventana, #5424569001) at 91 °C for 48 min. Primary antibody cocktails were applied after incubation with a human FC-receptor-blocking reagent (1:50; Miltenyi, #130-059-901).

The antibody cocktails for Staining set 1 included for the first incubation antibodies against glucagon (1:200; Abcam plc. #Ab10988) and tubulin beta 3 (1:25; R&D Systems, MAB1195), for the second incubation against mouse IgG1 con. AF555 (1:200; Invitrogen, #A-21127) and mouse IgG2a con. AF647 (1:50; Invitrogen, #A21241), and for the third incubation against somatostatin con. AF750 (1:200; Novus Biologicals, #NBP2-99309 AF750) with DAPI (1 ng/ml; Invitrogen, #D1306). The cocktails for Staining set 2 included for the first incubation antibodies against perilipin 1/PLIN1 (1:50; Progen, #690156) and PECAM1/CD31 (1:33; Abcam, #ab134168), for the second incubation against mouse IgG1 con. AF555 (1:100; Invitrogen, #A21127) and rabbit IgG con. AF750 (1:50; Invitrogen, #A21039), and for the third incubation against insulin con. AF488 (1:200; Invitrogen, #53-9769-82) with DAPI (1 ng/ml; Invitrogen, #D1306). WSI acquisition was performed with a slide scanner (Zeiss AxioScan.Z1 equipped with ZEN 3.10 software of the Light Microscopy Facility, a Core Facility of the CMCB Technology Platform at TU Dresden.) at ×20 magnification. Exemplary WSIs are shown in Supplementary Fig. 8.

### Deep learning models

**Model architectures.** Training DL models on WSIs requires special precautions due to the large file size. Since an entire image does not fit into GPU (graphics processing unit) memory, a patch-based approach is required, where patches are similar-sized crops of pixel values of tissue regions in the WSI. A difficulty to this approach is that only a global label per image is available, but not per patch. It is not known if the manifestation of the respective image-level class is actually given in every patch or if there are also patches that are not indicative of the image-level class and would hence confuse the models during training. A Multiple Instance Learning Approach (MIL) is commonly used to tackle such problems[23]. In MIL the individual patches are first encoded by a pre-trained feature extractor and subsequently pooled and fed together to a MIL classification algorithm that is trained to predict the image-level class label. Our patch size was 256 × 256 pixels covering 115.2 μm of the WSI. We tested two different feature extractors on the IHC WSIs. The first is a Vision Transformer pre-trained on ImageNet21k[24], which consists of natural images, while the second is Phikon[66], a Vision Transformer pre-trained on over six-thousand histologic WSIs and therefore specifically tailored to the domain.

As the MIL algorithm, we used Clustering-constrained Attention Multiple Instance Learning (CLAM)[25] as well as Chowder[67]. Both architectures deal with the variability in patch importance in different ways. CLAM uses an attention mechanism to focus on the most relevant patches within a bag while simultaneously imposing clustering constraints to ensure that similar patches receive comparable attention weights, enhancing interpretability and performance. Specifically, each patch receives a learned attention score indicating its importance, unlike in transformers, where attention is calculated between patches. The final slide-level representation is obtained by weighting the patch features by their attention scores and aggregating them. We adapted the CLAM algorithm to further improve generalizability by only showing a varying random subset of patches of a WSI (usually 5%)

to the model during an epoch. Moreover, a cosine annealing learning rate scheduler as well as gradient accumulation were added. The Chowder architecture employs a modular and hierarchical approach to MIL, where instance-level features are aggregated through a series of convolutional and pooling layers to capture complex patterns and dependencies. It also incorporates importance sampling to dynamically select the most informative instances from each bag, thereby reducing computational load and enhancing the model's ability to focus on critical data points.

**Data representations.** Patches of IHC samples were encoded as any other natural image as RGB images (3 color channels), similar to natural images. In contrast, mIF immunostainings required further preprocessing, since in these cases each channel contains the intensity values of one of the respective stainings, which do not coincide with color channels. Each of the mIF WSIs has 4 channels, containing 3 of the staining set-specific stainings as well as DAPI, as a nuclear (DNA) label. We tested 3 different data representations on the mIF WSIs:

- *RGB* This representation treats each of the 3 stainings as RGB color channels and additionally overlays the DAPI channel as a grayscale image on each of the other 3 channels, resulting in an RGB image similar to what can be seen with a standard visualization software when showing all stainings simultaneously. This representation contains the co-occurrence of the stainings, i.e., the model can learn which stainings occur together in the WSI.
- *Channel-wise* The channel-wise representation treats each channel (staining) individually as a grayscale image. Consequently, each patch gets encoded 4 times during feature extraction. The resulting features (1-dimensional vectors) are then appended, leading to a representation where each channel is encoded in detail, but no co-occurrence information is preserved. The resulting feature vectors have the same feature embedding size (1024) as the other representations but yield 4x more features than the other two representations.
- *Channel-wise average* The channel-wise average representation is very similar to the channel-wise representation but aims to include the co-occurrence of stainings by averaging the 4 different feature vectors respective to their occurrence in the image while losing specific details of individual stainings.

**Training procedure.** We split the data consisting of 100 unique patients into a train (75 patients) and test (25 patients) set and performed the model development solely on the train data. For that, we used a cross-validation, where in each of 15 splits we randomly used 60 patients from the train set for training and the remaining 15 patients for validation. Due to the different distributions of T2D within the two cohorts, we applied balanced sampling during training only for Tübingen patients, while the Dresden cohort was already balanced.

As the target metric, we chose the Area Under the Receiver Operating Characteristic Curve (AUROC), since for the subsequent XAI steps it is more important to have a model that has a high separability and can generally distinguish between the two classes (T2D or no T2D) than having a model that works well with a specific cutoff as measured by the F1-Score or the Accuracy.

## Explainable AI

**Attention methods.** Initially, we focused on determining the significance of specific patches using the built-in attention mechanism of the classification head. Nevertheless, raw attention data predominantly indicate general importance rather than class-specific relevance. To address the latter, we applied Attention Layer-wise Relevance Propagation (Attention LRP)[26], which filters the attention values to highlight patches distinctly associated with either diabetic or non-diabetic status within the same WSI. The class-specific relevance $R$ is iteratively computed for each layer $l$. Here, $x^{(l)}$ is the input at layer $l$

with $j$ elements and $x^{(l-1)}$ is the input of the downstream layer with $i$ elements. $W_{ji}$ is the respective weight matrix of the layer. Non-linear functions like GELU[68] produce both positive and negative outputs. To account for this, relevance propagation can be adjusted by forming a subset of indices $s$ resulting in the following relevance propagation:

$$R_j^{(l)} = \sum_{\{i|(i,j)\in s\}} \frac{x_j W_{ji}}{\sum_{\{j'|(j',i)\in s\}} x_{j'} W_{j'i}} R_i^{(l-1)},$$
$$where\ s = \{(i,j)|x_j W_{ji} \geq 0\}. \tag{1}$$

For initialization $r^{(0)}$ is set to $r^{(0)} = 1_k$, where $k$ is the one-hot-encoded outcome class. Based on the relevance, the average relevance filtered attention matrix $\widehat{A}$ is computed for each transformer block $b$. To this end, the expectation across all attention heads $h$ of the Hadamard product between the relevance and the attention head's gradient $\nabla A^{(b)}$ is taken, considering only the positive relevance:

$$\widehat{A}^{(b)} = I + \mathbb{E}_h(\nabla A^{(b)} \odot R^{(l_b)})^+. \tag{2}$$

The final attention attribution output is then computed by the weighted attention relevance across all $B$ transformer blocks.

**Attribution methods.** Compared to attention methods, attribution methods are architecture agnostic and do not require attention mechanisms in the model. The simplest form of attribution is the gradient from an outcome class $y_j$ w.r.t. the input $x$. When only considering the positive gradients, this method is called Saliency[69]. However, raw gradients are prone to gradient shattering[70] and vanishing[71]. Therefore, we employed Integrated Gradients (IG)[29], a more robust technique that offers greater contextual information for each input pixel by incorporating gradients with respect to a baseline value (we chose a black pixel as the baseline). To this end, we compute a path integral between a baseline value $x_0$ and the true value $x_j$ of each of the $j$ input pixels:

$$IG_j(x, x_0, f) = (x_j - x_0) \int_{\alpha=0}^1 \frac{\partial f(x_0 + \alpha(x - x_0))}{\partial x_j} d\alpha. \tag{3}$$

However, the initial selection of a baseline value in IG can be ambiguous, and computing multiple path integrals across various baseline values may be inefficient. To address this, Expected Gradients (EG)[27] avoids this selection of a baseline value by averaging the integrated gradients over a distribution of baseline images $x_0 \sim D$ (often a training data distribution or noise):

$$EG_j(x, f) = \mathbb{E}_{x_0 \sim D, \alpha \sim U(0,1)} \left[ \frac{\partial f(x_0 + \alpha(x_j - x_0))}{\partial x_j} d\alpha \right]. \tag{4}$$

In application, this expectation is approximated via a mini-batch sampling approach for $x_0$ and $\alpha$.

More related to relevance computing methods, DeepLIFT[72] works by comparing the activation of each neuron to its reference activation and assigning contribution scores based on the difference. Defining the difference between a target neuron $t$ and the baseline $t_0$ as $\Delta t = t - t_0$ (we chose $t_0 = 0$), the summation-to-delta of contribution scores is established. Here, $\eta$ represents the downstream hidden layer neuron, $K$, the number of downstream neurons, and $C$, the single contribution score, which quantifies the impact of downstream neuron $\eta_i$ on the difference in $t$. Based on the contribution scores, the multiplier $m_{\Delta\eta\Delta t}$ is constructed, inspired by the partial derivative in the backpropagation algorithm:

$$m_{\Delta\eta_i\Delta t} = \frac{C_{\Delta\eta_i\Delta t}}{\Delta t}\ with\ \sum_{i=1}^K C_{\Delta\eta_i\Delta t} = \Delta t \tag{5}$$

Based on this multiplier, we measure the contribution scores of the input: $m_{\Delta x\Delta t}$. To make results more robust, we advance all

attribution methods by SmoothGrad[73]. SmoothGrad computes the heatmaps several times based on noisy input samples with Gaussian noise (with $\sigma = 1.0$ and $n = 10$ in our case) and subsequently averages the heatmaps. The heatmaps are smoothed with a Gaussian Kernel with $\sigma = 1.0$. We focus exclusively on positive attribution, emphasizing factors that support a specific outcome. Exemplary heatmaps of all methods are depicted in Supplementary Fig. 9.

### Histologic biomarker

**ROI segmentation.** To compute the identified biomarkers a segmentation map of the respective regions is needed. Due to the amount and size of the WSIs we used scribble annotation[74] to efficiently label 44 (out of 600) IHC WSIs with the labels background, tissue, adipocytes, fibrotic patterns, and islets. Scribble annotation allows for a very fast annotation because regions are not annotated densely (i.e., each pixel gets annotated) but rather only smaller lines (scribbles) are used to approximately label the respective regions. During model training, we used the ignore label in nnU-Net[75] to only update the model weights based on annotated scribbles while ignoring the remaining image. While annotation is based on scribbles, the model still returns a dense prediction, which makes this approach well-suited for segmenting the ROIs. Since the nature of scribbles only allows a quantitative evaluation of the respective annotated areas, a qualitative analysis of the resulting segmentation maps was conducted. After adjusting some annotations and retraining the nnU-Net, a satisfactory segmentation quality was reached.

**Quantifying attention.** The attention to specific ROIs for the IHC stainings was computed by summing up the raw attention within specific segmentation masks and standardizing it by the total attention in the WSI and the tissue size. We summed up the attention per staining channel for the mIF stainings and standardized it by the total attention overall stainings and the tissue size. For the exploratory analysis, we z-standardized all continuous variables on non-interpretable scales; however, we routinely z-standardized all continuous covariates for the statistical analysis.

**Biomarker quantification.** For the IHC staining, we computed the area-related histologic biomarkers based on the segmentation masks and counted the number of distinct islets and adipocyte clusters by applying connected components to the respective segmentation mask. All biomarkers are standardized by the total size of the tissue. For the islet-related biomarkers, we only included islet segmentations larger than 5000 pixels, ignoring segmentation artifacts or errors. For the mIF-specific intensity biomarkers, we computed the channel-specific mean intensity as an approximation of the respective area of the trait and standardized it again by the tissue size. In the case of perilipin 1, we first threshold the intensity values, including only values larger than 1000 (intensity measured as 16-bit integers), filtering out non-specific stained traits.

### Statistical methods

**Hypothesis testing.** To test differences between two groups, e.g., patients with and without diabetes, we applied a two-sided two-sample t-test. To test if at least one group is significantly different, we applied the F-test within a one-way ANOVA. To test if a coefficient in a linear model is significantly different from zero we used the two-sided z-test. For all tests, we considered $p > 0.05$ as not significant.

**Statistical models.** For statistical analysis of the biomarkers, we used a generalized mixed linear model (GMLM), as we have several dependence structures within the data. For the diabetes status as the outcome, we selected the canonical binomial link function. For the HOMA2B level, we selected the canonical Gaussian and Gamma link functions and reported results for the Gaussian link, as its Akaike Information Criterion (AIC) was lower. We accounted for random effects linked to the cohort and staining method. Because there was no variation between the outcomes (i.e., diabetes status) within a person, we could not model the person as a random effect. In this case, the intercept would be enough to fit one model per patient perfectly. All models were fitted via maximum likelihood and the L-BFGS-B optimizer[76]. We tested for multicollinearity by computing variance inflation factor (VIF) and removed "benign tumor" due to perfect multicollinearity with "malignant tumor" combined with "chronic pancreatitis".

**Statistical model evaluation.** For all models with the same modality and response variable, we used the AIC for model selection. Additionally, we evaluated the logistic models through a confusion matrix and the linear models through parity plots, residual plots, and Q-Q plots of the residuals. All evaluation results of the statistical models are shown in Supplementary Figs. 10 and 11.

### Reporting summary

Further information on research design is available in the Nature Portfolio Reporting Summary linked to this article.

## Data availability

All data supporting the findings described in this manuscript are available in the article and in the Supplementary Information and from the corresponding author upon request. Source data are provided with this paper. The data to recreate the figures, as well as the statistical analysis within the paper, are uploaded as supplementary files. The raw WSI data, including IHC and mIF WSIs as well as the deidentified tabular clinical patient data generated in this study, have been deposited in the DZD database and can be downloaded with instructions on how to use them here: https://s.dzd-ev.org/diadem. The deidentified tabular clinical patient data includes: Patient_ID_Publication, Diabetic_Status, Sex, BMI_Categories (kg/m$^2$), HbA1c_Categories (%), Duration_of_diabetes_range, Fasting_Glucose_Categories (mmol/l), Fasting_Insulin (pmol/l), HOMA2B_(c-peptide)_categories, and Underlying_disease_corrected. The personalized tabular clinical patient data are available under restricted access for patient privacy; access can be obtained by sending an inquiry to one of the corresponding authors. No further related documents will be made public. The data will remain publicly available indefinitely following publication. The raw WSI data and deidentified tabular clinical patient data are openly accessible for research and non-commercial purposes. Use of the dataset requires agreement to the Creative Commons Attribution-NonCommercial 4.0 International (CC BY-NC 4.0) license. Source data are provided with this paper.

## Code availability

All code used to preprocess the data, train the deep learning models, apply XAI methods and to create the source data for the figures can be downloaded here: https://github.com/MIC-DKFZ/diabetes-xai.

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

## Acknowledgements

We thank all the participants of the "LIDOPACO" programs in Tübingen and Dresden. The studies were supported by the German Center for Diabetes Research (Deutsches Zentrum für Diabetesforschung, DZD). The DZD is funded by the German Federal Ministry for Education and Research and the states where its partner institutions are located (01GI0925). The authors acknowledge the project-specific financial support of the Helmholtz Association (project DIADEM, ZT-1-PF-5 139). This project has received funding from the European Union's Horizon Europe research and innovation program under grant agreement No 1010954433 (Intercept-T2D). Views and opinions expressed are, however, those of the author(s) only and do not necessarily reflect those of the European Union. Neither the European Union nor the granting authority can be held responsible for them. Special thanks to Rebekka Wehner, Susanne Doms, and Marc Schmitz (Institute for Immunology of the University Hospital Carl Gustav Carus Dresden) for supporting the development of the multiplex immunofluorescence staining. This work was funded by Helmholtz Imaging (HI), a platform of the Helmholtz Incubator on Information and Data Science. Birkenfeld A. was supported by the German Federal Ministry for Education and Research (01GI0925) via the German Center for Diabetes Research (DZD eV); Ministry of Science, Research and the Arts Baden-Württemberg; and Helmholtz Munich. The authors thank Darya Trofimova, Lars Krämer and Carsten Lüth of the DKFZ for insightful discussions and feedback. This work was supported by the Light Microscopy Facility, a Core Facility of the CMCB Technology Platform at TU Dresden.

## Author contributions

The Authors contributed in the following categories: Conceived and designed the experiments (L.K., S.Z., F.G., Y.M., P.J., F.I., M.S., and R.W.) Performed the experiments (L.K., S.Z., F.G., and Y.M.) Analyzed the data (L.K., S.Z., F.G., and Y.M.) Contributed materials/analysis tool (L.K., S.Z., F.G., Y.M., K.G., E.S., M.H., N.K., D.F., A.S., E.G., H.Y., H.H., S.W., S.N., A.K., A.M., D.H., F.F., D.A., J.W., R.J., M.D., K.M., A.B., S.U., P.J., F.I., M.S., and R.W.) Wrote the paper (L.K., S.Z., F.G., Y.M., P.J., F.I., M.S., and R.W.).

## Funding

## Competing interests

The Authors declare no competing interests.

## Additional information

[1]IML Group, German Cancer Research Center (DKFZ), Heidelberg, Germany. [2]Institute for Machine Learning, ETH Zürich, Zürich, Switzerland. [3]Helmholtz Imaging, German Cancer Research Center (DKFZ), Heidelberg, Germany. [4]Division of Medical Image Computing, German Cancer Research Center (DKFZ), Heidelberg, Germany. [5]Institute for Diabetes Research and Metabolic Diseases of the Helmholtz Center Munich (IDM), University of Tübingen, Tübingen, Germany. [6]Internal Medicine IV, Endocrinology, Diabetology and Nephrology, University Hospital Tübingen, Tübingen, Germany. [7]German Center for Diabetes Research (DZD e.V.), Neuherberg, Germany. [8]Department of Molecular Diabetology, University Hospital Carl Gustav Carus, Medical Faculty, Technische Universität Dresden, Dresden, Germany. [9]Paul Langerhans Institute Dresden (PLID) of the Helmholtz Center Munich, University Hospital Carl Gustav Carus and Faculty of Medicine of the TU Dresden, Dresden, Germany. [10]Internal Medicine I, Endocrinology and Diabetology, University Hospital Ulm, Ulm, Germany. [11]Institute for Clinical Chemistry and Pathobiochemistry, Department for Diagnostic Laboratory Medicine, University Hospital Tübingen, Tübingen, Germany. [12]Department of Visceral, Thoracic and Vascular Surgery, University Hospital Carl Gustav Carus, Medical Faculty, Technische Universität Dresden, Dresden, Germany. [13]Center for Molecular and Cellular Bioengineering, Technische Universität Dresden, Light Microscopy Facility, Dresden, Germany. [14]Section of Clinical Epidemiology, Department of Community Medicine, Kyoto University, Kyoto, Japan. [15]Center for Innovative Research for Communities and Clinical Excellence (CiRC2LE), Fukushima Medical University, Fukushima, Japan. [16]Department of General, Visceral and Transplant Surgery, University Hospital Tübingen, Tübingen, Germany. [17]Institute of Pathology and Neuropathology, University Hospital Tübingen, Tübingen, Germany. [18]Department of Pathology, University Hospital Carl Gustav Carus, Medical Faculty, Technische Universität Dresden, Dresden, Germany. [19]Pattern Analysis and Learning Group, Department of Radiation Oncology, Heidelberg University Hospital, Heidelberg, Germany. [20]Department of Endocrinology and Diabetology, Medical Faculty and University Hospital Düsseldorf, Heinrich Heine University Düsseldorf, Düsseldorf, Germany. [21]Institute for Clinical Diabetology, German Diabetes Center, Leibniz Center for Diabetes Research at Heinrich Heine University Düsseldorf, Düsseldorf, Germany. [22]Present address: Now at EPFL, Lausanne, Switzerland. [23]Present address: Now at Google DeepMind, London, UK. [24]These authors contributed equally: Lukas Klein, Sebastian Ziegler, Felicia Gerst, Yanni Morgenroth. [25]These authors jointly supervised this work: Paul F. Jäger, Fabian Isensee, Michele Solimena, Robert Wagner.
✉e-mail: paulfjaeger@icloud.com; f.isensee@dkfz.de; michele.solimena@tu-dresden.de; robert.wagner@uni-duesseldorf.de

