## [Transparent Peer Review File · Nature Communications]

Explainable AI-based analysis of human pancreas sections identifies traits of type 2 diabetes

Corresponding Author: Professor Robert Wagner

Version 1:

Reviewer comments:

Reviewer #1

(Remarks to the Author)

Overall, I believe the manuscript has improved significantly and has addressed many of my previous questions and concerns. I now understand most of the rationales and logic of the paper and am generally satisfied with their arguments. However, I still have some confusion regarding certain figures and paragraphs in the paper that need to be explained more precisely before its publication.

1) In the legend of Figure 3, the authors stated they used * to indicate actual diabetes states. However, * appears only in front of "No diabetes" in both panels A and B, suggesting that these samples are non-diabetic. Paradoxically, the panel marked with √, which indicates areas supporting a T2D diagnosis, seems to overlap with the regions labeled with x for "No diabetes." Therefore, I don't understand what this figure means. Additionally, it is essential to include controls showing attention heatmaps from samples with diabetes to achieve a complete understanding.

2) In the same sense, I don't know what "left side" refers to in the sentence: "increased attention to areas abundant in connective tissue (left side of the PECAM1-, glucagon-, insulin-, and somatostatin-stained WSIs) as well as a pronounced focus on adipocytes (top left of the perilipin 1-stained WSI). For readers to better understand the differences, I also suggest that the authors include enlarged ROIs to illustrate the detailed structures of the connective tissues and adipocytes mentioned.

3) Compared to T2D samples, non-T2D samples showed higher attention scores for islets in insulin-stained WSIs, whereas the opposite was observed for islets in somatostatin-stained WSIs (Fig. 4A). However, in Fig. 5B, the attention to islets labeled with somatostatin or insulin did not differ between T2D and non-T2D samples. In contrast, attention to islets labeled with glucagon showed significant differences. I wonder what causes these discrepancies.

(Remarks on code availability)

Reviewer #2

(Remarks to the Author)

Explainable AI (XAI) is indeed important in biomedical research, but it may not be the most appropriate tool for this particular problem. This work aims to identify histological features in pancreatic tissue that could predict type 2 diabetes (T2D), focusing on both the pancreas's exocrine tissue and the pancreatic islets. However, T2D prediction is neither based on nor diagnosed through histological sections, as the pancreas cannot be biopsied. What biological insights does the model provide to deepen our understanding of new disease biomarkers for treating or reversing type 2 diabetes? In addition, the authors made a substantial effort to revise the manuscript; however, several specific comments are below.

1. It is concerning that the chromogen versus fluorescence-based method for islet detection yielded such different results regarding islet number and size on consecutive paraffin-embedded sections, particularly when considering diabetes status. The data generated from fluorescence-based islet detection aligns with recent work demonstrating that the mass of beta,

alpha, and delta cells, the three primary types of islet endocrine cells, is unchanged in early-stage type 2 diabetes.

2. The most significant advantages of XAI are its ability to enhance model trustworthiness and reduce model complexity. Specifically, attribution scores can help verify that a model is making decisions based on relevant features rather than being misled by confounding factors. However, in this study, these advantages are not applicable since the predictive model used is unlikely to be widely adopted by other researchers - predicting disease states from images does not seem to be a standard or practical approach.

3. As the authors claim, XAI confirms hypotheses and lays the groundwork for future research. However, attribution and attention scores are complicated and need subjective human interpretation. The examined features in this manuscript still ultimately relate to channel-level analyses and correlate with conventional features (e.g., islet area). There are no clear examples of novel hypotheses generated by XAI in this work, particularly ones that would be difficult to analyze using traditional methods.

4. While the computation of attribution scores may be unbiased, the study's novel findings and its demonstration of XAI's power appear limited.

(Remarks on code availability)

Reviewer #3

(Remarks to the Author)

(Remarks on code availability)

Reviewer #4

(Remarks to the Author)

(Remarks on code availability)

Version 2:

Reviewer comments:

Reviewer #1

(Remarks to the Author)

Overall, I believe that the authors have adequately addressed my previous questions. I think the work is ready for the publication in Nature Communications.

(Remarks on code availability)

Reviewer #5

(Remarks to the Author)

In the present study, the authors develop an artificial intelligence-based tool for the analysis of histological sections of the pancreas from both healthy individuals and patients with T2D. The authors trained the model using a cohort of samples subjected to both chromogenic and fluorescent stainings and validated the training on a separate cohort. This approach enables the prediction of the patient's status (T2D or non-T2D), as well as the identification of other potentially disease-determining factors in an entirely unbiased manner, since artificial intelligence does not rely on the same visual criteria as the human eye to detect regions of interest and can identify patterns that may initially appear irrelevant.

My role in this review is to evaluate the authors' replies to the comments and concerns originally raised by referee 2, who was unable to provide a subsequent response. This evaluation is conducted through a detailed analysis of each specific question and the corresponding answer provided by the authors, followed by my own assessment after each point, as well as a general evaluation of the revised manuscript.

Q1. The referee expressed concern regarding the discrepancy between the number of islets and the size of consecutive pancreatic sections that differed only in the detection technique used (chromogenic versus fluorescent).

Response: The authors addressed this appropriately by including a direct comparison between both staining techniques and the number and size of islets (Supplementary Figure 14), showing in 100 randomly sampled observations that no significant differences were found between these variables across staining methods. They also emphasised that none of these variables were significant for predicting the diabetic state.

In my view, the authors convincingly demonstrate that no such differences exist between these parameters.

Q2. One of the referee's concerns was the limited advantages offered by this model and the difficulty of standardising it.

Response: The authors clarified that their intention was never to use AI as a diagnostic tool for diabetes per se, but rather to train the AI to predict T2D status based solely on pancreatic tissue and subsequently use it to determine which features are important for the model to distinguish between T2D and non-T2D, thereby generating new insights.

I share referee 2's concern that this technique would be difficult to standardise for application in other laboratories. However, the authors' use of this approach as a tool for new knowledge generation, in my opinion, supports the validity of their proposal.

Q3. Another concern raised by referee 2 was that the study does not present clear examples of genuinely novel hypotheses derived from explainable AI (XAI) that could not have been addressed using traditional methods.

Response: The authors claim that beyond the markers used, and although many of the results reaffirm what could already be determined by conventional techniques, the manuscript should be considered a proof of concept demonstrating that XAI can serve as a method to identify new biomarkers relevant to T2D without relying on predefined hypotheses.

In my opinion, I concur with the referee that the "new" discoveries in this study are rather limited. However, I believe that the application of XAI to the analysis of pancreatic samples represents a relevant contribution to scientific progress, as it may accelerate histological interpretation and the drawing of conclusions, both by validating existing knowledge and by uncovering previously unexplored markers that have so far not been considered relevant.

Q4. Finally, the referee noted that although the analysis is conducted impartially through AI, the demonstration of the power of XAI to generate novel findings appears limited in this study.

Response: The authors highlight several new findings from the present work:

1. Most notably, insulin was not the most relevant marker for the prediction of T2D, an unexpected finding suggesting that β -cell compartments beyond insulin-containing granules may play a crucial role in disease pathogenesis.
2. The model's attention to tubulin-3 revealed alterations in islet innervation, a novel and relatively unexplored aspect of T2D.
3. Furthermore, fibrotic patterns and pancreatic fat infiltration were emphasised as key alterations in T2D.

Following my previous reasoning, I concur with the authors that these discoveries, arising from the collaboration between humans and artificial intelligence to uncover previously unknown pathobiological mechanisms, represent a significant methodological advance with important biological and clinical implications.

In conclusion, the authors have adequately addressed the concerns raised by referee 2, providing clear and well-substantiated responses supported by additional analyses and clarifications. They have also considerably improved both the figures and the discussion, enhancing the overall clarity and scientific quality of the manuscript. I consider the findings presented, as well as the development of the proposed artificial intelligence-based tool, to be potentially relevant and of scientific interest, offering methodological innovation and contributing valuable insights into the histopathological characterisation of type 2 diabetes.

(Remarks on code availability)

Rebuttal

Changes within the manuscript are indicated by blue font color.

Reviewer #1

Q1 *What are the advantages of XAI over a large-scale analysis on pre-defined traits? For example, if we pre-define a series of traits (e.g., intensity of marker X in islets, distance of marker X and marker Y), calculate all traits, and run statistical tests against diabetes states (also considering co-variants), can we also obtain the same key traits as Fig. 1D? It is unclear whether XAI offers higher efficiency or accuracy over traditional analytical methods.*

We appreciate the reviewer's insightful comments regarding the novelty of our approach. The key advantage of an XAI-driven discovery process lies in its hypothesis-free nature, which reduces bias inherent in traditional biomarker identification methods. In the process described by the reviewer, all extracted biomarkers must be predefined before the experiments, potentially constraining the discovery to known features and limiting novel insights.

In contrast, our XAI-based approach allows for the unbiased identification of relevant regions of interest (ROIs), enabling direct comparison between XAI-derived ROIs and those hypothesized a priori by human experts. Observing whether these regions overlap or diverge provides (crucial) valuable information not only about type-2 diabetes but also about the limitations and biases of existing experimental approaches.

We acknowledge that the novelty of our approach, particularly the distinct steps of human-XAI interaction in biomarker identification and comparison, was not fully elaborated in the original manuscript. To address this, we have now added a dedicated results subsection, "Unbiased identification and computation of histologic biomarkers" (line 304), and revised the introduction to more clearly highlight the innovative aspects of the XAI-driven discovery process.

We hope these clarifications and additions sufficiently address the reviewer's concerns.

Q2 *The manuscript does not thoroughly explain the biological significance of the attention results to biologists. For example, in Fig 5a, DAPI receives high attention – does it suggest general cell morphology also contributes to classification? In Fig 5b & 5c, what does high attention of tubulin beta 3 imply? Should biologists focus on the spatial distribution of these markers or their interactions with other markers/cell types?*

We appreciate the reviewer's feedback and acknowledge that the concept and implications of attention and attribution may not have been sufficiently explained for researchers without an AI background. Attribution methods in general determine how much each input feature *attributes* to a model's output (i.e. how important each feature is in a prediction) through e.g. gradients or perturbations. Attention methods, a subset of attribution methods, leverage the output of the attention module used e.g. in the transformer architecture. Raw attention outputs indicate features of general importance to the model but lack a direct connection to specific outputs, requiring filtering (e.g., via gradients or relevance scoring) to highlight features used to predict a specific outcome.

In our approach, attention modules operate at the patch level rather than directly on individual pixels, meaning attention-based methods can only highlight the importance of each patch. To achieve a finer-grained analysis at the pixel level, we must rely on alternative attribution methods that can assess feature importance within each patch. As the reviewer correctly points out, high attribution to a specific feature—such as a particular cell type—indicates that the feature was utilized for prediction but does not inherently explain *why* it was important.

To address this, we went beyond classical attention analysis by applying novel strategies, such as summing attention within the DAPI channel in fluorescence microscopy to compare the relative importance of different multiplex channels. Additionally, we filtered the attention using Layer-wise Relevance Propagation (LRP) to assess the significance of each channel in relation to specific predicted outcome classes, such as diabetes status.

In our framework, attention and attribution maps serve as an unbiased filter, guiding the discovery of spatial features that are most relevant in the vast dataset. From these identified regions of interest, we derive hypotheses about potential biomarkers. For instance, we observed that the multiplex channel for tubulin beta 3 contributes significantly to the model's classification of patients with T2 diabetes. Based on this finding, we hypothesized that alterations in tubulin beta 3-stained structures, particularly within the islets, may be relevant. Consequently, we computed the biomarker "Tubulin beta 3 intensity within islets" to quantify this observation.

To better clarify the connection between attention results and specific biomarkers, and to provide a more specific interpretation of the attention maps, we have expanded the manuscript with a new results subsection, "Unbiased identification and computation of histologic biomarkers" (line 304). Further, the definition and differences between attribution and attention is now indicated in line 213 and 315.

We believe these additions improve clarity and ensure that the AI-driven analytical process is more accessible to a broader audience.

Q3 *The manuscript describes the technical necessity of training the model with subsets of the WSIs, named patches. What is the biological relevance of the selected patch size? How does it compare to the size of cells in the images?*

Since each pixel in our WSIs covered $\sim 0.45 \mu\text{m}$ and the patches covered 256×256 pixels, each patch had the biological size of $115.2 \mu\text{m}$ in height and width. Consequently, single patches contained multiple exo-/endocrine cells, each typically measuring $10\text{--}15 \mu\text{m}$ in diameter. In contrast, the diameter of the islets ranges from $50 \mu\text{m}$ for smaller islets to up to $200 \mu\text{m}$ for larger ones, while adipocytes can reach diameters of up to $100 \mu\text{m}$. Areas larger than the patch size are still seen by the model since in a later stage of the model (that is computationally less expensive and therefore feasible) the patches are all pooled together again before the final classification happens. Therefore, the model's decision is based on all information from the respective WSI and not only on a subset. We added this information to the Methods section (line 679).

Q4 *How was the islet segmentation performed? Figure 6 shows the relative difference in quantified XAI-derived histologic biomarkers between the control and T2D groups. The methods section describes quantifying these biomarkers (Lines 733 – 765). The ROI segmentation is dependent on the scribble annotation deep learning algorithm, which is a very recently described*

approach. Due to the novelty of this approach, particularly for application to pancreatic sections, the segmentation method should be validated appropriately. The manuscript lacks an explicit demonstration of successful islet segmentation. This is a concern, especially for sections with chromogenic stains that were not labeled for islet-specific markers.

The islet segmentation process was conducted using a standard dense labeling approach for semantic segmentation. To ensure comprehensive coverage and accuracy, we randomly selected 226 regions (each approximately 2000×2000 pixels, i.e. 900 x 900 μm, or 0.81 mm²) and manually annotated them. For the chromogenic WSI dataset, we employed a 5-fold cross-validation strategy, and the final predictions were obtained using an ensemble of all five segmentation models. The models achieved a mean Dice score of 0.8782 across all cross-validation folds, demonstrating the robustness of our approach.

For the fluorescent dataset, islet segmentation was performed completely manually. This was feasible because matching islet locations across multiplex channels made manual annotation more efficient compared to chromogenic slides, where each staining corresponds to a different tissue and WSI, increasing the annotation effort significantly.

Q5 *The description of donor demographics is unclear. Supplementary Table 2 describes the cohort donor metadata, but this table does not adequately describe important characteristics of the cohort. It is unclear how the cohort is split by sex and how many (integer) Controls/T2D patients are in each sex category. The reporting of Hba1c and other relevant physiological parameters is not split by diabetes status.*

We appreciate the reviewer's question; however, it appears that Figures 4, 5, and 6 in the Supplementary Material directly address these concerns. Specifically, Supplementary Figure 4A presents the demographic breakdown of the cohort, showing that it consists of 45 female and 55 male donors. Furthermore, it illustrates that 15 out of 45 female patients and 20 out of 55 male patients had T2D, which directly answers the reviewer's question.

Additionally, all relevant physiological parameters stratified by diabetes status are provided in Supplementary Figure 4, while demographic characteristics are detailed in Supplementary Figure 6.

Q6 *How do the histological biomarkers identified by XAI relate to disease duration? One would expect non-negligible effects on histological characteristics of pancreas tissue due to increased disease duration. Are predictive histological biomarkers consistent between sub-cohorts of differing disease duration?*

Initial results addressing this question are summarized in Supplementary Figure 12A, B, where correlation matrices reveal several significant but modest associations between disease duration and biomarkers in both chromogenic and fluorescence WSIs.

However, it is important to note that diabetes duration is a near-perfect predictor of diabetes status, except in cases where T2D was diagnosed for the first time from the biopsy. Specifically, all patients with a diabetes duration of zero are T2D-negative, while those with a nonzero duration are T2D-positive. Consequently, it is expected that biomarkers correlated with diabetes status also exhibit some correlation with diabetes duration. To assess whether these biomarkers correlate with disease duration independently of diabetes status, we performed a regression

analysis, incorporating both chromogenic and fluorescence biomarkers alongside diabetes status. The regression coefficients and p-values are summarized in the table below:

Brightfield ($duration_diabetes \sim diabetes_status + \{variable\}$):

Variable	Coefficient	p-value
Number of Islets	-0.1951	0.0117
Area of Islets	-0.1741	0.0008
Area of Adipocyte Clusters	0.2397	0.0098
Number of Adipocyte Clusters	0.0138	0.8589
Min. Dist. Islets and Adipocytes	-0.0913	0.3005
Area of Connective Tissue	-0.0876	0.3036

Fluorescence ($duration_diabetes \sim diabetes_status + \{variable\}$):

Variable	Coefficient	p-value
Number of Islets	-0.1079	0.2041
Area of Islets	-0.1599	0.0527
Tubulin beta 3 Intensity within Islets	0.0984	0.2359
Perilipin 1 Intensity	0.1089	0.1893

Notably, we observe significant negative correlations for Area of Islets (Chromogenic: -0.1741; $p = 0.0008$, Fluorescence: -0.1599; $p = 0.0527$) and Number of Islets (-0.1951; $p = 0.0117$), as well as a significant positive correlation for Area of Adipocyte Clusters (0.2397; $p = 0.0098$). These findings align with the general progression of T2D, where pancreatic islet structures are reduced over time. However, the increase in adipocyte cluster area is also commonly observed with aging, making it a potential confounding factor.

Overall, given the variability in disease progression among individuals, structural changes in pancreatic tissue may not always correlate with disease duration in a straightforward manner. Thanks to the reviewers advice and the relevance of the resulting findings, we now addressed this topic in the results (line 393) and discussion section (line 511). Further, we advanced the correlation matrices in Suppl. Figure 12A, B which is now colored based on the significance levels.

Q7 "(Line 153) We employed a 15-fold cross-validation, i.e., 15 different training ($n=60$ each) and validation ($n=15$ each) splits to avoid overfitting". The authors used an incorrect definition of K-fold validation. Typically, 15-fold validation means the data is split into 15 groups ($n=5$ each in this case) and one group is used as the test set in each run.

We acknowledge the reviewer's observation that we did not strictly follow the definition of splitting into 15 equally sized folds. As the reviewer correctly calculated, this approach would have resulted in validation folds containing only 5 cases, which we deemed too small for reliable validation. To address this, we instead created 15 different splits, ensuring that each validation set contained 15 cases, thereby improving the reliability of the validation process (see also reply to Q1 by Reviewer #3). To prevent any confusion, we have revised the wording in the manuscript to explicitly clarify this approach (line 158).

Q8 *While patch-level attention and attribution reveal the differential patterns between control and T2D samples, pixel-level attribution needs to be more conclusive. Beyond Figure 4, is it possible to find differential patterns of marker distribution (e.g., colocalization of glucagon and somatostatin) within the images when comparing T2D and control samples?*

We appreciate the reviewer's feedback and agree that the transition from slide-level heatmaps to specific biomarkers was presented too abruptly, making it challenging to follow. To address this, we have introduced a new subsection, "Unbiased identification and computation of histologic biomarkers" (line 304), where we explicitly interpret the specific cellular and multicellular structures that the model attends to and explain how we infer the respective biomarkers from these interpretations.

For instance, in Figure 6(d), as the reviewer hypothesized, we observe that the model indeed attends to the colocalization of fluorescent glucagon and somatostatin markers within pancreatic islets. We further quantify these findings for subjects with T2D and in the control group in two sections: the subsection "Quantifying attention to regions-of-interest," analyzes the differences in attention, and the subsection "Statistical analysis reveals a significant association of histologic biomarkers with diabetes status and insulin secretion," evaluates the differences between the resulting biomarkers.

These additions ensure a clearer, more structured transition from heatmaps to biomarker derivation, enhancing the interpretability of our approach.

Q9 *How well do attention/attribution models capture regions with lower marker intensities? And how well do they disregard regions with high but less informative marker intensities?*

We appreciate the reviewer's insightful question, which intertwines two key aspects: (1) the model's ability to detect lower marker intensities and (2) the role of attention and attribution methods in visualizing the features the model is sensitive to.

The model training process inherently determines whether certain regions are captured based on whether they provide a meaningful signal for solving the classification task. Once the model has learned which image features are relevant for the distinction between T2D and no diabetes, the reasoning of the model is visualized by the attention and attribution methods. These methods are not biased by the marker intensities but solely show how important the model deems certain features for its task. For instance, Figure 6(d) illustrates that even weak tubulin beta 3 staining is highlighted by the model, confirming that the model can attend to subtle but relevant features.

Given the importance of this discussion, we have incorporated these observations into the manuscript to further clarify how the heatmaps relate to the underlying model (line 220).

Q10 *Could authors provide more details on Multiple Instance Learning (MIL) in the model, particularly how attention is calculated? Is it the same as self-attention in Transformer? Is attention calculated between patches or from each patch to the final outputs?*

The Multiple Instance Learning (MIL) approach in the model follows the widely used Clustering-constrained Attention Multiple Instance Learning (CLAM) framework. Unlike naive MIL approaches, which often base its decision on a single patch, CLAM employs attention-based pooling to aggregate all patch-level features to make a prediction based on the full representation of the slide.

Thus, the attention computation in CLAM is not the same as self-attention in Transformers. Instead of computing relationships between all patches as in a Transformer, CLAM assigns an attention score to each patch based on its relevance to the classification task. These scores determine the contribution of each patch to the final slide-level representation.

As attention is calculated per patch, each patch receives an attention score indicating its importance. The final slide-level representation is obtained by weighting the patch features by their attention scores and aggregating them. Thus, attention is allocated to each patch, rather than between patches, and the relationship between the patches is learned through the downstream fully connected layers in the MIL classifier.

Based on the reviewers feedback, we highlight this difference to the application of attention in transformer architectures now in the method section (Line 689). For further information on CLAM, we refer to the respective paper [1].

[1] Lu, Ming Y., et al. "Data-efficient and weakly supervised computational pathology on whole-slide images." *Nature biomedical engineering*.

Q11 *The authors compared different methods of handling fluorescence WSI channels and claimed "(Line 182-183) the latter benefited from co-occurrence information reflecting the spatial relationships between different staining". Does concatenating the encodings along another axis (resulting in $Pn \times 4096$ encodings) preserve the co-occurrence relationship? In addition, the reason that the "channel-wise" model performs worse than the "channel-wise average" model might also be overfitting due to a higher dimensionality.*

We appreciate the reviewer's feedback and apologize for not explaining the handling of different fluorescence channels better. In our approach, we did not concatenate the feature vectors of the same patch; instead, we appended them as if they were independent patches. As a result, the model had no information about which feature vectors originated from the same patch. This ensured that, while the amount of patches is four times larger for the channel-wise representation, the feature length stays the same between all representations. This is important as the attention mechanism aggregates across the number of patches, and the final classification module only operates on one final vector that has the size of the feature length not influenced by the number of patches. This feature length is the same across all representations.

Furthermore, our results showed that the channel-wise average representation performed better for one staining set but worse for the other. Together with our extensive cross validation and the

test set of 25% of the patients, these results clearly indicate that different levels of overfitting due to image representation were not present (see also reply to Q1 by Reviewer #3).

To clarify this in the manuscript, we have revised the methods section (Line 722) to ensure that our approach is more clearly described.

Reviewer #3

Q1 *I understand the challenges involved in obtaining pancreatic sections from living donors, with 35 samples from T2D patients and 65 from non-T2D patients who underwent pancreatectomy for various pancreatic disorders. However, I believe that simply labeling six well-known proteins and applying a DL algorithm to such a small sample size is excessive. With such a limited sample size, overfitting is almost inevitable. Additionally, the potential for individual variability to dominate the sample reduces the likelihood of discovering significant new findings.*

We appreciate the reviewer's concern regarding the sample size and the potential for overfitting. While the patient number in this study might appear small for a DL project, it is one of the most extensive ones in the domain of diabetes microscopy. Further, we emphasize that the gigapixel-sized WSIs, including different specific markers and two microscopy modalities, provide a very diverse set of information even for one patient, as one WSI resulted on average in 4424 patches (i.e. around 3.5M patches in total).

In addition, this work represents, to our knowledge, the first DL-based approach for classifying T2D using human pancreatic tissue WSIs from living donors. Given the absence of prior studies in this area, we carefully selected proteins and labels that are well-established in T2D pathology to ensure that the AI model learns from clinically relevant histological markers, thereby strengthening the robustness of our findings.

We recognize the challenges posed by a small sample size and took several measures to mitigate overfitting and account for inter-individual variability:

1. Cross-validation strategy: Instead of conventional 15-fold cross-validation, which would have resulted in validation sets that were too small, we created 15 individual train/validation splits, each with 15 patients in the validation set. This approach ensures more reliable validation (see also reply to Q7 by Reviewer #1)
2. Bias reduction in performance metrics: Rather than averaging the performance across individual splits, we saved the predictions from all splits and computed the final metrics at the end. This prevents bias from outliers in any single split and provides more reliable validation metrics for hyperparameter tuning.
3. Independent test set evaluation: The final model was evaluated on a completely unseen test set of 25 patients. If significant overfitting had occurred, we would have observed a drastic drop in performance on these cases, which was not the case (see also reply to Q11 by Reviewer #1).
4. Phikon foundation model: We also used the Phikon foundation model as a patch encoder, which was pretrained on 40M histologic patches. The idea behind

foundation models is that they are pre-trained on a large volume of data and then fine-tuned for specific downstream tasks, leveraging their broad knowledge and adaptability to achieve high performance with minimal task-specific training.

Additionally, while the study is based on a limited number of 100 unique patients, this is counterbalanced by the rich dataset comprising multiple stainings and imaging modalities (chromogenic and fluorescence). Instead of pooling all patient data into a single model, we conducted independent experiments for each staining+modality combination, training separate models for each. This diverse experimental setup significantly reduces the likelihood that all models would overfit to the same features, as they were trained on different data (even if derived from the same patient).

Given that our conclusions are based on consistent trends across multiple independently trained models, rather than any single potentially overfitted model, we are confident that our findings remain valid despite the small sample size. To further clarify these methodological considerations, we have revised the manuscript accordingly.

Q2 *Considering these points, I am perplexed about the rationale behind undertaking such a complex procedure to obtain human pancreatic slices with minimal labeling, only to then use DL to extract differences of limited confidence. I disagree with the authors' claim that the current paper has identified "reliable biomarkers with reliable associations with T2D." Furthermore, the study does not underpin "key findings about pancreatic tissue alterations in T2D and provides novel targets for research", as the authors have suggested.*

As outlined in our response to Q1, we took careful steps to ensure the reliability of our DL models. Importantly, all biological conclusions were based exclusively on test set cases—meaning that all attention and attribution heatmaps originate from the test set, not the training set. This approach significantly reduces the risk of interpreting potentially overfitted or unreliable DL results.

Furthermore, as detailed in Q1, all biological interpretations are supported by multiple independent experiments, each involving separately trained DL models and statistical models on different datasets and biomarkers. The statistical models underwent extensive evaluation (see Table 1, Supplementary Figures 10 & 11, and Supplementary Table 3), demonstrating a strong association between the computed biomarkers and diabetes status.

While we acknowledge that overfitting can never be entirely ruled out in any deep learning study, we have carefully designed our experiments to minimize this risk and limit our interpretations strictly to the examined dataset. Given that our findings are consistently supported across multiple independent analyses, we are confident in their reliability and robustness.

We believe that we have identified multiple important, even if subtle, morphological aspects related to the T2D pathology. Our analysis reveals that pancreatic adipocytes are located closer to islets in samples from patients with T2D. While this might initially seem like a minor finding, it holds significant implications due to the role of the pancreatic microenvironment in islet function. This proximity likely influences the exocrine-endocrine crosstalk, which is critical for the islets. Most interesting, a previous study reported that "fatty infiltration in the human pancreas creates a neurotrophic microenvironment and promotes remodeling of pancreatic innervation" [1]. Given

that both adipocytes and beta cells, i.e. fatty acid release via lipolysis and insulin secretion, are strongly regulated by neuronal signals, we believe that our observation opens a new avenue for exploring the pathogenesis of diabetes. By linking these observations, we offer a new perspective on the complex interaction between pancreatic fat, islet function and neuronal regulation. This leads us to another key finding: the best performance in diabetes classification was achieved for the neuronal marker tubulin beta 3 in the chromogenic WSI, and for staining set 1 (labeling neurons, alpha, and delta cells) in the fluorescence modality. This suggests that structural changes in the neuronal network as well as in alpha and/or delta cells—changes that are not yet fully understood—may not only accompany but also actively contribute to the progression of the disease. We address these findings in more detail in our discussion part (line 441, 447, and 562).

[1] Tang, Shiue-Cheng, et al. "Human pancreatic neuro-insular network in health and fatty infiltration." *Diabetologia*.

Q3 *In fact, quantifying attention to regions of interest for chromogenic WSIs (Figure 4) and fluorescence samples (Figure 5) from non-diabetic and T2D donors yielded inconsistent results. For the chromogenic WSIs, only minimal differences were observed in PECAM1, Perilipin 1, and tubulin in some cases, while no differences were detected in insulin, glucagon, or somatostatin. I believe this indicates a failure or insensitivity of the method in confirming well-known biomarkers. In contrast, the fluorescence samples showed some differences in glucagon as well as tubulin, PECAM1, and Perilipin 1. Therefore, I do not understand the authors' assertion that "islets are attended above average for tubulin beta 3 and PECAM1 and below average for Perilipin 1 staining compared to the rest of the tissue (Fig. 5C)."*

We appreciate the reviewer's feedback but believe there is a fundamental misunderstanding regarding the presented results. The passage cited by the reviewer about chromogenic WSIs pertains specifically to *attention to connective tissue*, not the general importance of stainings. These results assess how relevant connective tissue is for each of the six models trained on different stainings, given that none of these stainings are specific to connective tissue.

Our standardized attention scores confirm that attention to connective tissue is consistent across all six stainings, which is expected. Furthermore, we observe that connective tissue becomes more important when classifying a patient as T2D positive, with a significant difference for PECAM1, Perilipin 1, and tubulin beta 3. This is reasonable, as the non-significant stainings specifically target islets, where attention is significantly higher compared to non-islet-specific stainings. The referred results in the fluorescence analyses then focus on *attention to total tissue and specific marker channels*, which are fundamentally different and not directly comparable to the referred chromogenic WSI results.

To improve clarity and prevent misinterpretation, we have revised the manuscript by modifying Figure 4 for better readability and refining the corresponding text in the section.

Q4 *The identified differences in the DL network's attentions for PECAM1, Perilipin 1, and tubulin were later determined to be related to the sizes of the cells involved. I do not consider this to be a novel finding, and I do not see the necessity of going through such extensive efforts to uncover such an apparent morphological feature. Technically, this may be due to the fact that the authors divided the whole WSI into smaller patches to extract features. This*

approach tends to focus solely on obvious local features and may miss relatively large, distributed features that DL algorithms could detect, but human observers might not.

We believe there is again a fundamental misunderstanding of our results. We are also unsure how the reviewer arrived at the conclusion that *“the identified differences in the DL network’s attentions for PECAM1, Perilipin 1, and tubulin were later determined to be related to the sizes of the cells involved,”* as this is not stated in the manuscript.

For example, differences in attention to tubulin beta 3 in fluorescence-stained WSIs—particularly when comparing attention within islets versus the whole tissue—motivated the computation of the biomarker “tubulin intensity within islets” as a proxy for the number of nerve cells in islets. Cell size is just one of many biomarkers computed across different cell types in both chromogenic and fluorescent-stained WSIs, but none of the cell size markers is specific for nerve cells (tubulin beta 3) or blood vessels (PECAM1).

Additionally, the assertion that our findings are due to patching of WSIs is incorrect. While each patch receives one attention score, the MIL classifier processes all patches from a WSI simultaneously, enabling it to detect larger ROIs and critical spatial features—a well-established approach in computational histopathology [1,2]. Furthermore, Figure 3 (LRP-based attention heatmaps) clearly shows that the model identifies large ROIs, such as adipocyte clusters and connective tissue structures spanning across the tissue.

Overall, we believe that reviewer #3 has misinterpreted both the methodology and the presented results. To improve clarity and to prevent this from happening again, we have added an additional section explicitly explaining the connection between attention heatmaps and biomarker computation.

[1] Lu, Ming Y., et al. “Data-efficient and weakly supervised computational pathology on whole-slide images.” *Nature biomedical engineering*.

[2] Ilse, Maximilian, Jakub Tomczak, and Max Welling. “Attention-based deep multiple instance learning.” *International conference on machine learning*.

Q5 *Overall, I do not accept the assertion that “The AI-based T2D prediction models demonstrated a highly reliable performance,” and I do not believe that the study represents a carefully designed experiment or a significant technical advancement.*

While we appreciate the reviewer’s opinion, we respectfully disagree with her/his assessment, as her/his questions and statements indicate a fundamental misunderstanding of both the methodology and results. Additionally, some referenced results were taken out of context, and certain claims were made regarding findings that are not present in the manuscript.

To our knowledge, this study is the first to classify T2D directly from human pancreatic tissue while also introducing a novel approach for unbiased biomarker detection. The rigorous experimental design, including extensive cross-validation and independent test set evaluation, demonstrates the high predictive performance of our models across two microscopy modalities as described in our extensive answer to Q1.

Beyond the results, the entire experimental pipeline was carefully designed—from patient selection and the choice of clinically relevant histological markers to DL model training, XAI application, and statistical evaluation. Furthermore, this study goes beyond conventional experiment pipelines, as each step involves interaction between human experts and AI models, making it a pioneering approach in scientific discovery.

Given these considerations, we strongly disagree with the reviewer's conclusion. The novelty of our work is evident not only in the results but also in the innovative experimental process used to achieve them.

Answer to all Reviewers

We sincerely thank all reviewers for their valuable comments. We appreciate the general consensus among the reviewers regarding the significance of our work for the community and the improvement since the last revision: “[...] I am generally satisfied with their arguments” (“Reviewer 1”), “[...] the authors made a substantial effort to revise the manuscript” (“Reviewer 2”).

We addressed all concerns in our point-by-point responses to each reviewer and revised the main paper while retaining the overall structures. Based on the reviewers' feedback, we have made the following improvements to the manuscript (location in manuscript in parentheses):

1. Improving Figure 3 by highlighting WSI regions discussed in the text and showing an enlarged section below (Page 8).
2. Adding another Figure similar to Figure 3 but with the WSIs of a T2D-positive patient (Suppl. Figure 13, Page 38).
3. Adding a Figure showing the distribution of the number of islets and average islets size computed based on the fluorescence segmentations only (Suppl. Figure 14, Page 39)
4. Improved explanation of the limitations of attention heatmaps and how we include the scientist in the loop (Line 226).
5. Specifying discussed ROIs attended by the model in more detail (Lines 240, 246, and 348).
6. Highlight more the discrepancy between modalities when interpreting results (Lines 276 and 281)
7. Strengthen the point that the novelty lies also in the methodology (Lines 118 and 640)
8. Highlighting that XAI is used for scientific discovery in this work and not solely for model trustworthiness (Line 104)
9. Discussing the results with a more in-depth focus on new findings (Lines 560, 598, 627, and 632)

Answers for Reviewer 1

Q1: In the legend of Figure 3, the authors stated they used * to indicate actual diabetes states. However, * appears only in front of "No diabetes" in both panels A and B, suggesting that these samples are non-diabetic. Paradoxically, the panel marked with \checkmark , which indicates areas supporting a T2D diagnosis, seems to overlap with the regions labeled with x for "No diabetes." Therefore, I don't understand what this figure means. Additionally, it is essential to include controls showing attention heatmaps from samples with diabetes to achieve a complete understanding.

Answer: We thank the reviewer for bringing up the seemingly paradoxical fact that heatmap areas indicating high attention of the model when predicting T2D can overlap with areas the model attends to when predicting no T2D. This is expected as the attention maps only show

what *spatial areas* are important for each predicted outcome, not the specific traits within these areas. In our setting, it is up to the biological interpretation, leveraging prior human knowledge, and further analysis of the heatmaps to determine these traits.

For example, the size of a cell can support one outcome but its shape another one. In both cases, the same area, i.e. the cell, is of importance but due to different traits. To further clarify this, we added an explanation in Line 226 to the manuscript.

Moreover, we agree that it is essential to add respective heatmaps of a control case. We focus on one case in the main manuscript to be able to show all different modalities (6 individual chromogenic stainings as well as both fluorescence staining sets). Consequently, we added a new figure representing a control case to the supplementary material (page 40).

Q2: In the same sense, I don't know what "left side" refers to in the sentence: "increased attention to areas abundant in connective tissue (left side of the PECAM1-, glucagon-, insulin-, and somatostatin-stained WSIs) as well as a pronounced focus on adipocytes (top left of the perilipin 1-stained WSI). For readers to better understand the differences, I also suggest that the authors include enlarged ROIs to illustrate the detailed structures of the connective tissues and adipocytes mentioned.

Answer: We agree that Figure 3 needs further descriptions of the underlying WSI structure that is shown as an example in this figure. We adjusted the figure so that the original WSIs without heatmaps are enlarged and specific areas such as fibrotic patterns or adipocytes are better visible.

We have clarified the mentioned sentence to specify that "left side" refers to the left side of the original WSIs rather than the figure orientation in Lines 246-249. Beyond the enlarged ROIs available in Figure 6, we also enhanced Figure 3 by adding bounding boxes to highlight the specific areas referenced in the manuscript text. In addition, we also show the enlarged content of the bounding boxes now below the WSIs.

Q3: Compared to T2D samples, non-T2D samples showed higher attention scores for islets in insulin-stained WSIs, whereas the opposite was observed for islets in somatostatin-stained WSIs (Fig. 4A). However, in Fig. 5B, the attention to islets labeled with somatostatin or insulin did not differ between T2D and non-T2D samples. In contrast, attention to islets labeled with glucagon showed significant differences. I wonder what causes these discrepancies.

Answer: We appreciate the reviewer's question and the chance to clarify these findings. First, it is important to mention that the differences in the attention to islets for insulin and somatostatin-stained chromogenic WSIs (Fig. 4A) are not statistically significant, similar to the fluorescence equivalents (Fig. 5B). For glucagon, we see a difference in the fluorescence staining.

This discrepancy may originate from the difference between the modalities. The fluorescence modality obtains information from multiple stainings in the model at the same time (as it is a staining set). Therefore, one possible explanation could be that glucagon is only relevant when interacting with the other two stainings from that staining set

(somatostatin and tubulin) but not by itself. In addition, chromogenic stainings have a dynamic range of only 1-2 orders of magnitude compared to 3-5 orders of magnitude in the case of immunofluorescence stainings. The optimization of the multiplex fluorescence protocols required the use of different antibodies, for instance against glucagon (mouse monoclonal, 1:200; Abcam plc. #Ab10988, mouse), that those used for chromogenic stainings (anti-glucagon rabbit polyclonal, 1:600; Santa Cruz, #sc13091). We highlight this discrepancy between the modalities when interpreting the results now more in Lines 276-285.

Answers for Reviewer 2

Q1: It is concerning that the chromogen versus fluorescence-based method for islet detection yielded such different results regarding islet number and size on consecutive paraffin-embedded sections, particularly when considering diabetes status. The data generated from fluorescence-based islet detection aligns with recent work demonstrating that the mass of beta, alpha, and delta cells, the three primary types of islet endocrine cells, is unchanged in early-stage type 2 diabetes.

Answer: We appreciate the reviewer's concerns about the islet segmentation methods applied to the chromogenic and fluorescence WSIs. While we use a deep learning-based semantic segmentation on the chromogenic WSIs and a manual annotation on the fluorescence WSIs, the results between both modalities are very similar (excluding Perilipin as the segmentation model failed here). When only visualizing the distribution of the standardized islet size and number of islets for the fluorescence modality (Suppl. Figure 14), we observe that their distribution is very similar to that of the other chromogenic stainings, again excluding Perilipin. An additional t-test supports this observation. As we agree that this is an important question we added the distribution of the average islet number and size, including the statistical tests, to the Appendix page 40 as Supplementary Figure 14.

The average islet size detected in the fluorescence WSI is smaller in T2D patients compared to ND individuals, supporting the results of the chromogenic staining presented in Fig. 7A. While most published studies report no significant differences in islet size or number, i.e. beta cell mass between early-stage T2D and ND patients, prolonged metabolic stress may contribute to the reduction in beta cell mass in long-lasting diabetes patients [1]. As our cohort is heterogeneous in terms of diabetes duration including also individuals with long-lasting diabetes (Supplem. Table 2), this might explain the observed differences.

However, our statistical models (Table 1a for chromogenic and Suppl. Table 3a for fluorescence), as well as the correlation analysis (Suppl. Figure 12), demonstrate that islet area and number of islets exhibit comparable effects across both modalities, showing similar coefficient signs and magnitudes. Notably, neither parameter reaches statistical significance in predicting T2D status.

[1] Chen, C., Cohrs, C. M., Stertmann, J., Bozsak, R., & Speier, S. (2017). Human beta cell mass and function in diabetes: Recent advances in knowledge and technologies to understand disease pathogenesis. *Molecular metabolism*, 6(9), 943-957.

Q2: The most significant advantages of XAI are its ability to enhance model trustworthiness and reduce model complexity. Specifically, attribution scores can help verify that a model is making decisions based on relevant features rather than being misled by confounding factors. However, in this study, these advantages are not applicable since the predictive model used is unlikely to be widely adopted by other researchers - predicting disease states from images does not seem to be a standard or practical approach.

Answer: While one of the advantages of XAI is indeed increasing the trustworthiness of a model, as the reviewer correctly noted, the main goal is different in our study. We are not training models to be able to diagnose T2D from pancreas tissues since this is much easier done using blood samples. Instead, we aim to use XAI to gain a better understanding of the disease itself. Using XAI to understand data is the purpose of an emerging subfield of XAI for knowledge or scientific discovery, where scientists use XAI to understand complex and high-dimensional data from domains such as semiconductor design [2], solar cell production [3], or antibiotics discovery [4].

As discovering spatial traits of pancreatic tissue related to T2D is a nontrivial objective, we hypothesized that this approach could also work with our data and problem. To this end, we train models that can predict the T2D status solely based on the pancreatic tissue, a task that cannot be performed by a human, and then use XAI to see what features are important for the model to distinguish between T2D and non-T2D. By interpreting these features, new insights can be generated.

We have revised the manuscript to better clarify this important distinction in our XAI methodology and now provide a more detailed statement in Line 104.

[2] Choubisa, H., Todorović, P., Pina, J. M., Parmar, D. H., Li, Z., Voznyy, O., ... & Sargent, E. H. (2023). Interpretable discovery of semiconductors with machine learning. *npj Computational Materials*, 9(1), 117.

[3] Klein, L., Ziegler, S., Laufer, F., Debus, C., Götz, M., Maier-Hein, K., ... & Jäger, P. F. (2024). Discovering process dynamics for scalable perovskite solar cell manufacturing with explainable AI. *Advanced Materials*, 36(7), 2307160.

[4] Wong, F., Zheng, E. J., Valeri, J. A., Donghia, N. M., Anahtar, M. N., Omori, S., ... & Collins, J. J. (2024). Discovery of a structural class of antibiotics with explainable deep learning. *Nature*, 626(7997), 177-185.

Q3: As the authors claim, XAI confirms hypotheses and lays the groundwork for future research. However, attribution and attention scores are complicated and need subjective human interpretation. The examined features in this manuscript still ultimately relate to channel-level analyses and correlate with conventional features (e.g., islet area). There are no clear examples of novel hypotheses generated by XAI in this work, particularly ones that would be difficult to analyze using traditional methods.

Answer: We respectfully disagree with this assessment. Our approach is fundamentally novel in reversing the traditional biomarker discovery paradigm: rather than starting with human assumptions, we systematically analyze which features the AI model prioritizes, filtering potential biomarkers without initial bias before human interpretation.

The convergence of XAI-derived features with conventional markers represents validation, not limitation. Two fundamentally different methodologies independently identifying similar

biomarkers provides mutual confirmation and demonstrates that our models learn biologically meaningful patterns rather than artifacts.

Our methodology strategically combines AI's strength in processing large-scale datasets and detecting subtle patterns with human expertise in biological interpretation and abstraction. This hybrid approach enables the discovery of features that purely traditional, hypothesis-driven methods might overlook, particularly complex spatial relationships across large tissue areas.

The correlation with known features establishes biological plausibility while our novel methodology opens pathways for feature discovery that traditional approaches can often not achieve. To strengthen this point that the novelty lies also in the methodology we added a sentence in Lines 118 and 640.

Q4: While the computation of attribution scores may be unbiased, the study's novel findings and its demonstration of XAI's power appear limited.

Answer: Our study addresses a fundamental methodological challenge in biomedical research: the analysis of large-scale and multi-modal histological datasets that have only a very limited tractability using traditional approaches. The volume and complexity of multi-modal pancreatic tissue data would be prohibitively expensive and time-consuming to analyze comprehensively through conventional manual annotation methods.

Our approach represents the first human-AI collaborative framework for pancreatic histology-based biomarker discovery in diabetes research, where XAI and deep learning provide a more unbiased computational filter to identify critical regions of interest from vast histological datasets. This methodology reduces human bias in feature selection while maintaining biological interpretability through XAI techniques.

This novel approach revealed several biological insights that either support or challenge current assumptions in T2D pathophysiology. Most remarkably, insulin was not the most relevant marker for T2D prediction, which is a counterintuitive finding suggesting that beta cell compartments beyond insulin-containing granules may be critically relevant for disease pathogenesis. The model's attention to tubulin uncovered alterations in islet innervation, a novel and poorly understood T2D aspect that provides a rationale for investigating parasympathetic synapses on beta cell primary cilia [5].

Additionally, our findings highlight fibrosis patterns and pancreatic fat infiltration as key alterations in T2D, which could have clinical implications in diabetes prevention and treatment, by directing future research to address fibrotic processes and reduce pancreatic fat accumulation.

These discoveries exemplify the transformative potential of human-AI collaboration in uncovering previously unknown pathobiological mechanisms at scale, representing a significant methodological advancement with substantial biological and clinical implications.

[5] Müller, A., Klena, N., Pang, S., Garcia, L. E. G., Topcheva, O., Aurrecochea Duran, S., ... & Solimena, M. (2024). Structure, interaction and nervous connectivity of beta cell primary cilia. *Nature Communications*, 15(1), 9168.